# High-time resolved radon-progeny measurements in the Arctic region (Svalbard Islands, Norway): results and potentialities

Roberto Salzano[1], Antonello Pasini[2], Antonietta Ianniello[2], Mauro Mazzola[3], Rita Traversi[4], Roberto Udisti[4]

[1]Institute for Atmospheric Pollution Research, National Research Council of Italy, Sesto Fiorentino (FI), Italy
[2]Institute for Atmospheric Pollution Research, National Research Council of Italy, Monterotondo (RM), Italy
[3]Institute of Atmospheric Sciences and Climate, National Research Council of Italy, Bologna (BO), Italy
[4]Department of Chemistry "Ugo Schiff", University of Florence, Sesto Fiorentino (FI), Italy

*Correspondence to*: Roberto Salzano (roberto.salzano@cnr.it)

**Abstract.** The estimation of radon progeny in the Arctic region represents a scientific challenge due to the required low limit of detection in consideration of the limited radon emanation associated with permafrost dynamics. This preliminary study highlighted, for the first time above 70°N, the possibility to monitor radon progeny in the Arctic region with a higher time resolution. The composition of the radon progeny offered the opportunity to identify air masses dominated by long-range transport, in presence or not of near-constant radon progeny instead of long and short lived progenies. Furthermore, the different ratio between radon and thoron progenies evidenced the contributions of local emissions and atmospheric stability. Two different emanation periods were defined in accordance to the permafrost dynamics at the ground and several accumulation windows were recognized coherently to the meteo-climatic conditions occurring at the study site.

## 1 Introduction

The detection of radionuclides within the Arctic environment is an important tool to help understanding the pathways for radionuclide transport to, within and from the Arctic (Chun, 2014; AMAP, 2010). Naturally-occurring radionuclides, emitted by geologic sources and associated with cosmogenic processes, can describe air-masses origin and residence time (Baskaran, 2016). This is a key information for studying the fate of pollutants in the Arctic region, which is controlled by the meteo-climatic conditions occurring in the different seasons and in the different days of the year (Baskaran and Shaw, 2001). From a seasonal point of view, the extension of the so-called *"Arctic front"* (Stohl, 2006) can deflect, in fact, air masses originated in continental areas (such as Northern Europe, Russia, Asia and North America) to higher altitudes, reducing the contribute to the deposition processes. Radon ($^{222}$Rn and $^{220}$Rn have half-lives, respectively, of about 3.8 days and 55 s) and its progeny represent an important tracer of the meteo-climatic conditions occurring in the lower atmosphere. The use of naturally-occurring nuclides, and in particular of radon, for pollution purposes was extensively investigated at lower latitudes (Duenas et al., 1996; Perrino et al., 2001; Sesana et al., 2003; Chambers et al., 2011; 2015), especially in urban settings. These case studies support the scientific community to use $^{222}$Rn as a comparatively simple and economical approach for defining the

stability conditions of the lower troposphere and for estimating the mixing height (Pasini and Ameli, 2003; Sesana et al., 2003; Veleva et al., 2010; Griffith et al., 2013; Pasini et al., 2014; Salzano et al., 2016). The low emissive conditions of the ground, controlled by the permafrost dynamics, limit the application of this approach in Polar Regions. The expected radon activities in the air (we refer to especially to 222Rn that is more frequently estimated in literature) ranges in the Arctic between 30 mBq

m$^{-3}$, with persistent polar winds, and more than 400 mBq m$^{-3}$ when continental air masses reached higher latitudes (Samuelson et al., 1986). This value is, of course, influenced by the latitude, by the meteo-climatic conditions, by the altitude of the sampling site and, finally, by the distance of continental areas. This range is coherent with Antarctica when "oceanic" air masses occurs and it is less stringent when "continental" or "local" air messes are incoming (Chambers et al., 2014). The occurrence of a melting season coupled with the higher extension of bared local and remote soils, potential sources of radon

emissions, let the requirement of significant low levels of detection (LLD) less stringent in Svalbard islands. The approach to the reduction of LLD can, in fact, be based on having high-volume sampling and/or high-sensitivity detectors (Chambers et al., 2014). The available techniques can be classified considering the half-life ($t_{1/2}$) of the considered isotopes. A first methodology is based on the direct measurement of radon nuclides ($^{222}$Rn or $^{220}$Rn) detecting the in-equilibrium progeny ($^{218}$Po, $^{214}$Po and $^{216}$Po, with $t_{1/2} < 20\ min$). Some indirect techniques include the detection of short-lived isotopes (such as $^{214}$Bi and

$^{214}$Pb, with $t_{1/2} < 1\ hour$) and of long-lived nuclides (such as $^{212}$Bi, $^{212}$Pb with $1 < t_{1/2} < 10\ hour$s). Finally, some indirect methods are based on the detection of near-constant progeny (such as $^{210}$Pb and $^{210}$Bi, with $t_{1/2} > 1\ day$). The main advantage of the direct measurement consists in the formation under controlled (aerosol-free) conditions where radon is in-equilibrium with its unattached progeny. Furthermore, the direct methods differ from the others by the introduction of delay volume necessary for the removal of not-in-equilibrium progeny. The physico-chemical behavior of radon is the key feature in all of

these techniques: it is a noble gas that, once emitted by soil, leaves the surface by molecular diffusion or by convection, and enters the atmosphere where the turbulent mixing diffuses nuclides (Porstendorfer, 1994). This gas is chemically unreactive and physi-sorption through electrostatic attraction on particles is negligible (Bocanegra and Hopke, 1988). On the other hand, the radon decay products are metallic elements that are easily physically fixed to existing aerosol particles in the atmosphere. Either the reduction of these particles in the atmosphere occurs by radioactive decay or by removal processes (dry deposition,

rainout, washout). Furthermore, turbulent mixing controls the distribution of this aerosol component in the troposphere. Different techniques allow estimating of this important tracer, all of them based on the physical behavior and/or the decay chain of this gas. The collection of $^{218}$Po by electrodeposition coupled to alpha-particle detection (Wada et al., 2010) represents the most common "*direct*" measurement of Rn. Furthermore, the lowest detection limit (below 10 mBq m$^{-3}$) can be obtained isolating the gaseous phase, removing the $^{220}$Rn component and detecting the freshly-decayed metals (Griffith et al., 2016).

Looking at the "*indirect*" measurement of Rn on particulate matter, the most common approach is based on collecting and counting the total activity associated with the short-lived (S$_\beta$) and the long-lived (L$_\beta$) progenies (Paatero et al., 1994; Perrino et al., 2000; Levin et al., 2002; Salzano et al., 2016). The determination of the near-constant (C$_\beta$) progeny completes, in conclusion, this picture. The collection and the detection steps, in this case, can be separated and samples can be stored for a

significant time (Paatero et al., 2010). Considering advantages or disadvantages, the lowest detection limits can be obtained having a relatively more complex sampling/counting system that requires more space for the installation. The availability of an accurate instrument implies the requirement of reliable calibration procedures that in remote areas could be restricted by logistic reasons, as in our case. Furthermore, the "*direct*" systems cannot detect $^{220}$Rn decay product due to the impossibility

to separate contributions using alpha detectors. The "*indirect*" methods have, on the other hand, a reduced impact in terms of resources necessaries for the installation and for maintenance but they require the assumption of equilibrium between radon gas and its aerosol progeny. This assumption is generally considered suitable for sites that are at a significant distance from the radon terrestrial source, when the sampling height from the ground is significant, if weather conditions are fairly calm, but is likely to fail under high-humidity and severe-sea conditions. This study will focus the attention on the technique based on

the not-in-equilibrium progeny, where high-volume sampling is not required. This is a single-filter approach coupled with beta counting and it represents, now, the best compromise for our logistic resources between detection efficiency and required man intervention. Furthermore, the disequilibrium issue is less invasive compared with the most common approach available in literature about the Arctic region (Zhang et al., 2015), where near-constant progeny is involved.

The present study tests the potentiality to study hourly variations instead of daily samples. Different authors have already
investigated seasonal trends highlighting the role of the Arctic haze on radon and its progeny (Suzuki et al., 1996; Paatero et al., 2010; Zhang et al., 2015). We will describe the high-frequency behavior of radon progeny looking at the persistence of stability conditions and we will combine these results with the air-mass characterization based on back-trajectories.

## 2 Methods

The study was carried out at the Gruvebadet observatory, facility of the Arctic Station "Dirigibile Italia" located in Ny Ålesund,
Spitzbergen (79°N, 11°E, 50 m a.s.l.). The site (Fig. 1) is located in the Brøgger Peninsula that is NW-SE oriented in front of the Kongsfjorden. Different glacial valleys (Austre - Midtre - Vestre Lovénbreen, Austre Brøggerbreen, etc.) slope down from the reliefs where the highest altitude is 1017 m above the sea level. Additional facilities are available nearby for the characterization of the physical conditions of the lower atmospheric boundary layer. The survey covered the period from 1 April to 28 October 2015, including the melting season and the entire summer season.

**2.1 Radon progeny**

The natural radioactivity was measured using an automatic stability monitor (PBL Mixing Monitor, FAI Instruments, Fontenuova, Rome, Italy) with a sampling height of 3 m above the ground. The system comprises an air sampler for the collection of particulate matter on filter membranes and a Geiger-Muller counter for determining the total β activity of radionuclides attached to the particles. The instrument operates on two filters at the same time: while the sampling phase is
acting on one filter for 1 h, another filter is performing the β detection at four different intervals (0-10, 10-20, 30-40, 40-50 minutes). These instrumental features ensure that the β activity of the particles is continuously determined over an integration

time of 1 h and that the β measurement period is long enough to guarantee highly accurate results. The automatic subtraction of the background radiation (Perrino et al., 2000) improves the accuracy of the determination and the maximum instrumental error at the lowest counting level was about 8%. $T_\beta$ in Eq. (1) is the sum of β particles emitted by different nuclides sampled in the aerosol ($[N]_\beta^i$):

$T_\beta = \sum_{i=1}^{n}[N]_\beta^i$                      (1)

Each nuclide is collected on the filter after a sampling period and it is detected during a counting interval. Differential equations (Islam and Haque, 1994) regulate both phases and the sampling step can be generalized as:

$\frac{d[N]_s^i}{dt} = v[N]_{air}^i + \lambda_{i-1}[N]_s^{i-1} - \lambda_i[N]_s^i$                      (2)

The first term on the right side represents the collection obtained specifying the air sampling flow rate ($v$) in $m^3\ h^{-1}$ and activity

in the air ($[N]_{air}^i$). The second term defines the contribution of the eventually occurring parent isotope (i-1) on the filter and the third term is the decay component of the daughter nuclide (i). Those decay terms considers the specific decay constant of the parent ($\lambda_{i-1}$) and daughter ($\lambda_i$) isotopes. Furthermore, the presence of the nuclide in the air is described in Eq. (3) by the combination of the locally originated nuclide ($[N]_L^i$) added to transported contribute ($[N]_T^i$).

$[N]_{air}^i = [N]_L^i + [N]_T^i$                      (3)

Similar differential equations describe the counting phases and the daughter decay and the eventual supply of the parent nuclide control the β emission:

$\frac{d[N]_\beta^i}{dt} = \lambda_{i-1}[N]_s^{i-1} - \lambda_i[N]_s^i$                      (4)

Considering only naturally-occurring nuclides, Eq. (1) can be described by the sum of β emissions produced by $^{222}$Rn progeny ($S_\beta$), $^{220}$Rn progeny ($L_\beta$) and some near-constant nuclides including cosmogenic isotopes ($C_\beta$).

$T_\beta = [^{214}Pb]_\beta + [^{214}Bi]_\beta + [^{212}Pb]_\beta + [^{212}Bi]_\beta + C_\beta$               (5)

Excluding from $C_\beta$ the contribution of $^{210}$Pb, due to the low β energy emission ($E_\beta < 100\ keV$) where the detector has a very low efficiency (Lee and Burgess, 2014), the remaining near-constant nuclides are $^{210}$Bi ($t_{1/2} \sim 5\ days$ and $E_\beta \sim 1162\ keV$), $^{10}$Be ($t_{1/2} > 10^6\ years$ and $E_\beta \sim 556\ keV$) and $^{14}$C ($t_{1/2} \sim 5700\ years$ and $E_\beta \sim 156\ keV$). While the $^{14}$C contribution is limited by the low efficiency of detectors at low energies and by the limited amount of carbon present on filters (below 1 μg

$m^{-3}$), the $^{10}$Be component is limited by the low activities present in the atmosphere. Summarizing, we have one rapid-decay component ($S_\beta$ decreases 60 - 70 % within one hour) and one near-constant member ($C_\beta$). The intermediate term ($L_\beta$) reduces its activity to about 5 - 15 % after one hour. The mixing between those three components defines the final decay behavior observable at an hourly scale with four different counting steps. We can have two different seasonal behaviors in the Arctic region. The first one occurs especially during the Arctic winter, when the local emission of radon (both $^{222}$Rn and $^{220}$Rn) is

negligible ($L_\beta \simeq 0$) and the residence time of aerosol over sea (more than 2 days) is higher in presence of the so-called "Arctic haze" ($C_\beta > 0$). The second one occurs especially in the summer, when the local component is significant and the Arctic haze

is reduced ($S_\beta \gg L_\beta > C_\beta$). We assume under both conditions that transient equilibrium is occurring between the two progenies ($[^{214}Pb]_{air} = [^{214}Bi]_{air}$ and $[^{212}Pb]_{air} = [^{212}Bi]_{air}$). Some bias can occur especially during the summer when the local source is dominating over transport and the disequilibrium between progenies can be significant. The Bateman's solutions support the development of the above-mentioned differential equation Eq. (2) concerning the sampling phase:

$$[^{214}Bi]_S = 1.51[^{214}Pb]_S \tag{6a}$$

$$[^{214}Pb]_{air} = 1.97 \, v^{-1} [^{214}Pb]_S \tag{6b}$$

$$[^{212}Bi]_S = 1.02[^{212}Pb]_S \tag{6c}$$

$$[^{212}Pb]_{air} = 1.03 \, v^{-1} [^{212}Pb]_S \tag{6d}$$

$$C_\beta^{air} = v^{-1}C_\beta \tag{6e}$$

The Eq. (4) regarding the counting intervals must be solved for each period. The solution must consider the first and the last counting periods: from 0 to 10 minutes (first interval $T_\beta^1$) and from 40 to 50 minutes (forth interval $T_\beta^4$) after the end of the air sampling.

$$T_\beta^1 = \epsilon_{1024}[^{214}Pb]_S d_1^1 + \epsilon_{3272}[^{214}Bi]_S d_2^1 + \epsilon_{570}[^{212}Pb]_S d_3^1 + \epsilon_{2252}[^{212}Bi]_S d_4^1 + C_\beta \tag{7a}$$

$$T_\beta^4 = \epsilon_{1024}[^{214}Pb]_S d_1^4 + \epsilon_{3272}[^{214}Bi]_S d_2^4 + \epsilon_{570}[^{212}Pb]_S d_3^4 + \epsilon_{2252}[^{212}Bi]_S d_4^4 + C_\beta \tag{7b}$$

The coefficients in Eq. (7a and 7b) are the detector efficiencies at each energy ($\epsilon_{keV}$) and the decay parameters ($d_i^n$) obtained solving exponential equations (Eq. 4) for each i$^{th}$ isotope at each n$^{th}$ counting interval. We were not able to determine routinely the detector efficiency at each energy but it was possible to make some experiments with a similar instrument and some reference materials such as a KCl standard (we prepared a known $^{40}$K filter with $E_\beta \sim 1311 \, keV$) and a $^{137}$Cs-contaminated soil (we prepared a $^{137}$Cs-enriched filter with $E_\beta \sim 531 \, keV$ where the activity was determined by γ-spectrometry). This preliminary calibration requires a stronger effort for estimating precisely the efficiency at different energies but a relative ratio between the detector efficiency at 570, 1024 and 2252 keV normalized to the efficiency at 1024 keV was estimated in order to study the variations of the three β-emitting components ($S_\beta$, $L_\beta$ and $C_\beta$). We found that $\epsilon_{570} \sim 0.41\epsilon_{1024}$, $\epsilon_{1162} \sim 1.1\epsilon_{1024}$, $\epsilon_{2252} \sim 1.8\epsilon_{1024}$ and $\epsilon_{3272} \sim 2.1\epsilon_{1024}$. Substituting these parameters to Eq. (7a and 7b) and solving the system including a $^{220}$Rn to $^{222}$Rn ratio ($f$), we obtained:

$$S_\beta = 1.97 \frac{(N_{\beta,1}-N_{\beta,4})}{(2.15+0.74f)} \, v^{-1} \tag{8a}$$

$$L_\beta = 3.74f \frac{(N_{\beta,1}-N_{\beta,4})}{(2.15+0.74f)} \, v^{-1} \tag{8b}$$

$$C_\beta = [N_{\beta,4} \frac{(2.14f+1.89)}{(2.15+0.74f)} (N_{\beta,1} - N_{\beta,4})] \, v^{-1} \tag{8c}$$

All quantities are expressed in cps m$^{-3}$. Nevertheless, we prefer to enhance the contribution to the scientific community assuming the equilibrium between progenies during the observed period ($^{214}Pb_{eq} = S_\beta \, {}^{212}Pb_{eq} = L_\beta$). Assuming also that $\epsilon_{1024} \sim 10\%$, having in mind that further efforts are necessary for a reliable calibration, the comparison of our results with literature is possible obtaining activities expressed in mBq m$^{-3}$. Regarding C$_\beta$, the conversion requires a deeper knowledge

about this component and we prefer consequently to keep relative counts. The minimization of the chi-squared indicator, calculated between the four counting intervals and the respective values simulated between the two end-member situations ($C_\beta = 0$ or $L_\beta = 0$) supported the estimation of the three components. The optimization algorithm was developed in the R-Project programming environment (R Core Team, 2016). The lower limit of detection, in terms of $^{222}$Rn, of the stability monitor was estimated at 150 mBq m$^{-3}$ (Salzano et al, 2016) using an independent technique. Considering the logistic restrictions of the study site, routine quality check and sampling efficiency assessments were not possible. These limitations forced us to express $^{214}Pb_{eq}$ and $^{212}Pb_{eq}$ as mBq* m$^{-3}$ and to estimate $C_\beta$ as relative counts. The respective lowest limit of detections were 200 mBq* m$^{-3}$, 250 mBq* m$^{-3}$ and 0.015 cps m$^{-3}$, respectively.

## 2.2 Soil Rn-flux

The estimation of the soil Rn flux ($\phi$) was obtained using a stationary model where the major controlling factors are the soil radon emanation power and the soil water saturation (Zhuo et al., 2008). The model, based on Equation 10, required as input parameters the soil temperature ($T_S$), the soil water content (S), the soil Ra content ($R$), the soil density ($\rho_b$) and the soil porosity ($p$). In addition to some constants that are included, such as the radon decay constant ($\lambda$) and the diffusion coefficient of radon in the air ($D_0$), the emanation power ($\varepsilon$) can be calculated following the equations described by Zhuo et al. (2008).

$$\phi = R\rho_b\varepsilon(\frac{T_S}{273})^{0.75}\sqrt{\lambda D_0 p e^{-6Sp-6S^{14p}}} \qquad (9)$$

Compared to the preliminary description presented in Salzano et al. (2016), the description of the soil thaw depth is a critical input parameter in cold regions. Permafrost can be idealized as a two-layer system where the upper active layer overlays a frozen water-saturated layer. From this perspective, we approached the problem considering the 9 m temperature profile provided by the Bayelva borehole (Paulik et al., 2014), which supported also the estimation of the average temperature of the active layer. Under-prediction can affect the developed model under unsteady conditions since it is a simplified solution. Appropriate validating activities are required in order to evaluate the performance of this model. The estimation of the soil water content was approached using remotely sensed data provided by the EUMETSAT organization. We selected the soil moisture product obtained by the ASCAT sensor (Brocca et al., 2017), which is a real-aperture radar operating at 5.255 GHz (C-band) (EUMETSAT, 2015).

## 2.3 Back trajectories

The analysis of the back trajectories was approached calculating the air mass path with the HYSPLIT model (Stein et al., 2015). We considered 5 days trajectories using the GDAS meteorological dataset. Simulations were targeted on the study site at different altitudes (500 and 1000 m a.s.l.) in order to evaluate the circulation without the influence of orography on trajectories (Esau and Repina, 2012). This issue is extremely important considering the complexity of the studied fiord system and the extension of Svalbard island compared to the model resolution.

## 3 Results

Two different questions were approached in order to evaluate the potentialities in using radon-progeny in the Arctic region: what is the impact of permafrost dynamics on the radon detection in the air? How does the air mass trajectory control the signal detected in the lower atmosphere?

### 3.1 The contribution of the local soil flux to the air concentration

The evolution of the three radioactive components (Fig. 2 a, b and c) was the result of the overlapping of different sources and processes: local emission, oceanic income, slope flows and atmospheric stability. All of these factors can have different seasonal behaviors and this framework highlighted the need of a time-series analysis. The first step consisted, in fact, in de-trending the time series isolating the local-source component, which is controlled by the emissive condition of the local surface. The remaining seasonal components (distinct in low and high emanation periods) where analyzed removing the high-frequency bias (hourly oscillations) using a smoothing procedure based on a centered weighted moving average with a 24-hours window (Cowpertwait and Metcalfe, 2009). The main feature distinguishing the identified periods was the amplitude of variations concerning the natural radioactivity. While small fluctuations (up to 1 Bq* m$^{-3}$) in $^{214}$Pb$_{eq}$ were detected during the low-emanation period, $^{212}$Pb$_{eq}$ seemed to be very close the LLD and C$_\beta$ was frequently detected. The high-emanation period was characterized by sharp variations up to 8 Bq* m$^{-3}$ in terms of $^{214}$Pb$_{eq}$, by significant variations concerning $^{212}$Pb$_{eq}$ and by an occasionally detectable amount of C$_\beta$. The presence of $^{212}$Pb$_{eq}$ and C$_\beta$ showed also a specific behavior: while the long-lived progeny followed the trend described by short-lived component (increasing from 260 mBq* m$^{-3}$ in the first period to 400 mBq* m$^{-3}$ in the second one), C$_\beta$ showed the opposite (0.016 cps m$^{-3}$ in the first period and 0.015 cps m$^{-3}$ in the second one). Furthermore, looking at peaks defined by values higher than the third quartile, different episodes can be identified for each progeny component during each emanation period. While the $^{214}$Pb$_{eq}$ and the $^{212}$Pb$_{eq}$ components showed episodes that are significantly matching in terms of maxima, especially during the high-emanation period (r$^2$ = 0.79), C$_\beta$ showed peaks completely independent from the other components. The reconstruction of an indicative soil Rn-flux (Fig. 2d) supports the interpretation of the observed seasonality in the time-series. The two different emanation periods can be identified also looking at the soil Rn exhalation rate, which is controlled by the thermal behavior of the permafrost layer. The presented model output required two important input parameters (the thickness of the active layer and the average soil temperature) obtained from borehole measurements at the surface and from the remotely observed water saturation of the ground. The final output of the stationary emanation model indicated a limited soil emission until the end of May with a maximal emissive condition of local soils reached after 30 days at the beginning of July. Referring to the impact of permafrost dynamics, we can distinguish between a low-emanation period (which includes the ablation, the fusion and active-layer development phases) and a high-emanation period (as soon as the ground reached the maximal thickness of the emanating active-layer). The transition between the two periods can be positioned approximately in 8-9 July 2015 within a very short time interval. The observed abrupt impact of soil emanation on natural radioactivity in the air is coherent with the expected stabilization of soil exhalation obtained from the

model. This consistency implies that atmospheric processes are influenced by a regional source where emissive conditions of the ground are mostly stabilized. The model output is, in fact, representative only for the Ny Ålesund site as permafrost observations are site-specific (the study area is a coastal zone), and they cannot describe the overall behavior of more internal areas where orography is very complex.

## 3.2 The contribution of the meteo-climatic conditions to the air concentration

The combination between the three components could represent a diagnostic tool capable to describe the meteo-climatic conditions occurred on air masses. Looking at the occurrence of isotopic mixtures with different relative percentage of each component, the most frequent situation during both emanation periods was, of course, the condition dominated by the short-lived progeny, with a frequency ranged between 72% and 78%. The partition of contributions between the near-constant isotopes and the classes dominated by the long-lived progeny represented the most significant difference between the two periods. While the intermediate conditions between short-lived progeny and $C_\beta$ were more consistent during the low emanation period (respectively about 7% and 2%), the terms between short and long lived progenies were almost dominant during the second period (respectively about 15% and 26%). The first behavior is consistent with the end of the Arctic winter, when the Arctic haze enriches polar air masses with nuclides such as $^{210}$Pb and consequently $^{210}$Bi (Zhang et al., 2015). Furthermore, those conditions could highlight the occurrence of "*old*" and/or high-altitude air masses that were persistent at higher latitudes where radon sources are negligible. On the other hand, the presence of the long-lived progeny could trace the contribution of local sources or the arrival of recently emitted air masses (this component has a limited residence time in the atmosphere due to its complete decay reached after 50 hours). The importance of detecting these different components consists in the possibility to estimate the "*age*" and the "*origin*" of air masses (Chun, 2014). The relationship between each radon-progeny mixture and the wind features highlighted (Fig. 3) that the $^{214}$Pb$_{eq}$-$^{212}$Pb$_{eq}$ ratio was strictly controlled by the occurrence of wind calm conditions. This diagnostic ratio increases, in fact, in correspondence of stagnant conditions when win speed was lower than 4 m s$^{-1}$. This pattern was particularly evident during the summer period (Fig.3b) when the ratio is higher than 10 under calm conditions and it was less pronounced during the melting season (Fig.3a) with values between 5 to 12. The arrival of oceanic air masses (NW) and glacier-slope flows (NNE and ESE) contributed to reduce the ratio close to the LLDs ratio. Furthermore, it is important to notice that strong winds carried air masses with a lower content of radon progenies. This behavior was more significant during the melting season compared to the summer period and it implies that low wind-speed conditions favored the accumulation of nuclides (atmospheric stability) compared to advective situations that moved all the components to the specific lower limits of detection. This observation is in contrast with $C_\beta$, which can reach significant levels when advective conditions are associated only with glacier flows. Additional analyses were not possible at the moment since the near-constant component was very close to the estimated LLD and we can use this formation just as trigger of potential intrusion of higher tropospheric air masses. The study site is, in fact, deeply influenced by orographic effects and it can be described as a system that might be heavily stratified (Di Liberto et al., 2012; Esau and Repina, 2012; Mazzola et al., 2016). The role of wind-calm conditions was relevant if we consider that near-stable conditions of the lower atmosphere favor the accumulation of nuclides.

From this perspective, we can infer that $^{214}Pb_{eq}$ and $^{212}Pb_{eq}$ were coexistent when local emission and atmospheric stability are dominant compared with long-range transport. This last process can be identified when the long-lived is negligible ($^{212}Pb$ and $^{212}Bi$ is completely decayed after 50 hours) and air masses could be "*recent*" or "*aged*" in presence or not, respectively, of the short-lived progeny. This study cannot, at the moment, analyze these issues since the LLDs of the two minor components (long-lived and near-constant progenies) are still too high and a longer dataset is mandatory for providing a solid statistics. Nevertheless, back-trajectories and the observed absolute humidity provided additional information. The occurrence of periods characterized by high $^{214}Pb_{eq}$ - $^{212}Pb_{eq}$ ratios (above 4 and 8 during the different seasons) or by significant $C_\beta$ activities (Fig. 4a) were, in fact, associated with atmospheric stability and advection of flows arriving from the open sea or from the glaciers. The first driving factor is the persistence or not of winds below 4 m s$^{-1}$ and this distinction was controlled by atmospheric stability. It is important to remember, in this case, that while low wind speeds reduce the efficiency of local radon emanation, high wind speed contrasts accumulation of nuclides in the air but favors radon exhalation (see Sect. 2.2). Atmospheric stability can be traced looking at the residence time of air masses (estimated for 500 and 1000 m altitude backtrajectories) above Svalbard islands (we considered an area of 350 km radius) in the last 5 days (Fig. 4c). This information is not completely exhaustive since it does not consider the vertical intrusion of air masses through the inversion layer. The decoupling between the above-inversion and below-inversion air circulation in the fiord seemed to play an important role especially during the summer period when a long wind-calm window (about 20 days in August) occurred. During this event the radon-progeny ratio was permanently high but the residence time of air masses was very low (less than 12 hours with occasional spikes). The dynamics of the inversion layer in the fiord system will be investigated in the next future considering additional observations that can describe the fiord system better than a coarse-resolution model like HYSPLIT. The evolution of absolute humidity provided some confirmations about the importance of this process (Fig.4b). The general increasing trend over the whole campaign was, in fact, combined to accumulation windows, probably related to atmospheric stability, and to abrupt decrements (more than 1 g m-3) that were probably associated with glacier flows. These slope flows, with a katabatic behavior indicated also by high wind speed (more than 8 m s-1), were coincident with significant $C_\beta$ activities.

The availability of a longer dataset coupled to a reliable calibration of this technique are the first steps that must be addressed in order to obtain more a detailed description of complex processes in polar areas.

## 4 Conclusions

The detection of β emission from airborne particles with a high-time resolution offers the opportunity to increase the capability of studying atmospheric processes in polar areas. The reduction of soil exhalation during spring may appear as a limitation, but it represents an important challenge. The composition of the radon progeny in the Arctic region (above 70°N), defined for the first time with a high-time resolution, supported the identification of air masses dominated by long-range transport (with life up to 20 days and more than 20 days) in presence or not of near-constant radionuclides instead of long and short lived progenies. This study supports to extend this approach from the definition of the accumulation processes involving isotopes

present in the lower atmosphere, to the identification of the stability conditions of the lower atmosphere, to gather information about air masses and the soil-exhalation conditions. Two different emanation periods were defined in accordance to the permafrost occurrence at the ground. Furthermore, accumulation windows were recognized coherently to the meteo-climatic conditions occurring at the study site. This preliminary attempt must be continued with a longer time series in order to statistically analyze the correlation between radioactivity and mixing state of the lower atmosphere. However, we are confident that coupling this method with traditional chemical determinations on gases and aerosols, a more complete picture of pollutant dynamics in the Arctic region can be achieved.

## 5 Acknowledgments and Data

This study was supported by the logistic service provided by the Earth System Science and Environmental Technologies Department of the National Research Council of Italy. We would like to thank EUMETSAT for providing data products concerning the soil moisture content and AWI for the available dataset about the permafrost thermal profile. The authors gratefully acknowledge the NOAA Air Resources Laboratory (ARL), for the provision of the HYSPLIT transport and dispersion model used in this publication, and NCEP, for providing the Global Data Assimilation System (GDAS). This manuscript was language-proofed by Lena Rettori.

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

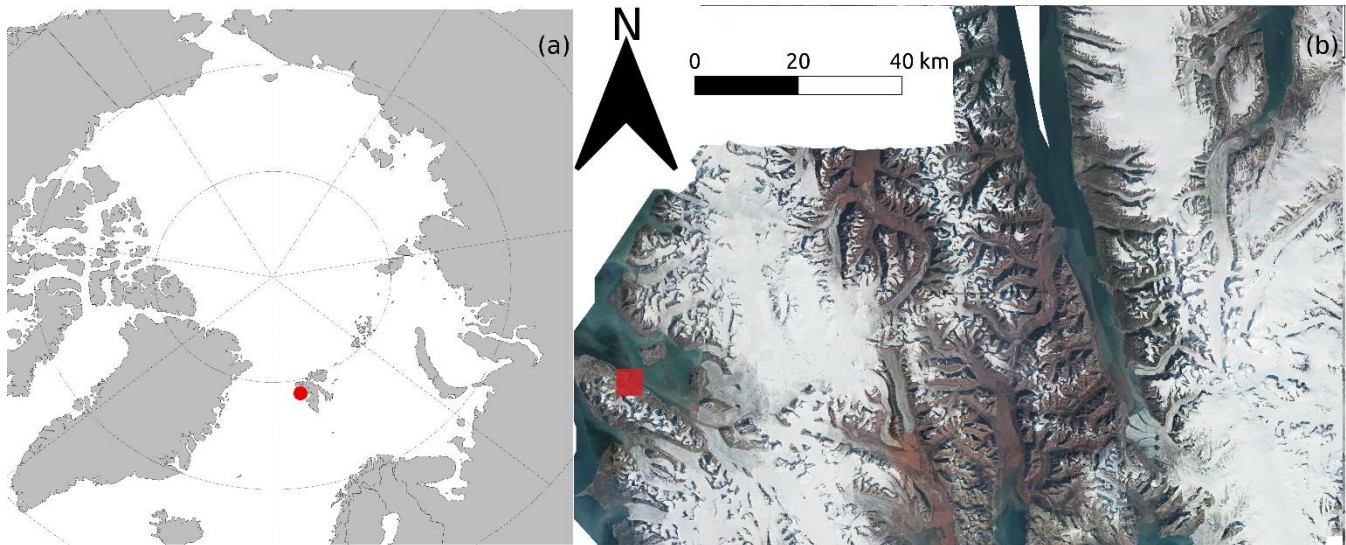

**Figure 1: Location map (a) of Ny Ålesund (Svalbard Islands, Norway) and zoom (b) on the Gruvebadet observatory (courtesy of Norwegian Polar Institute).**

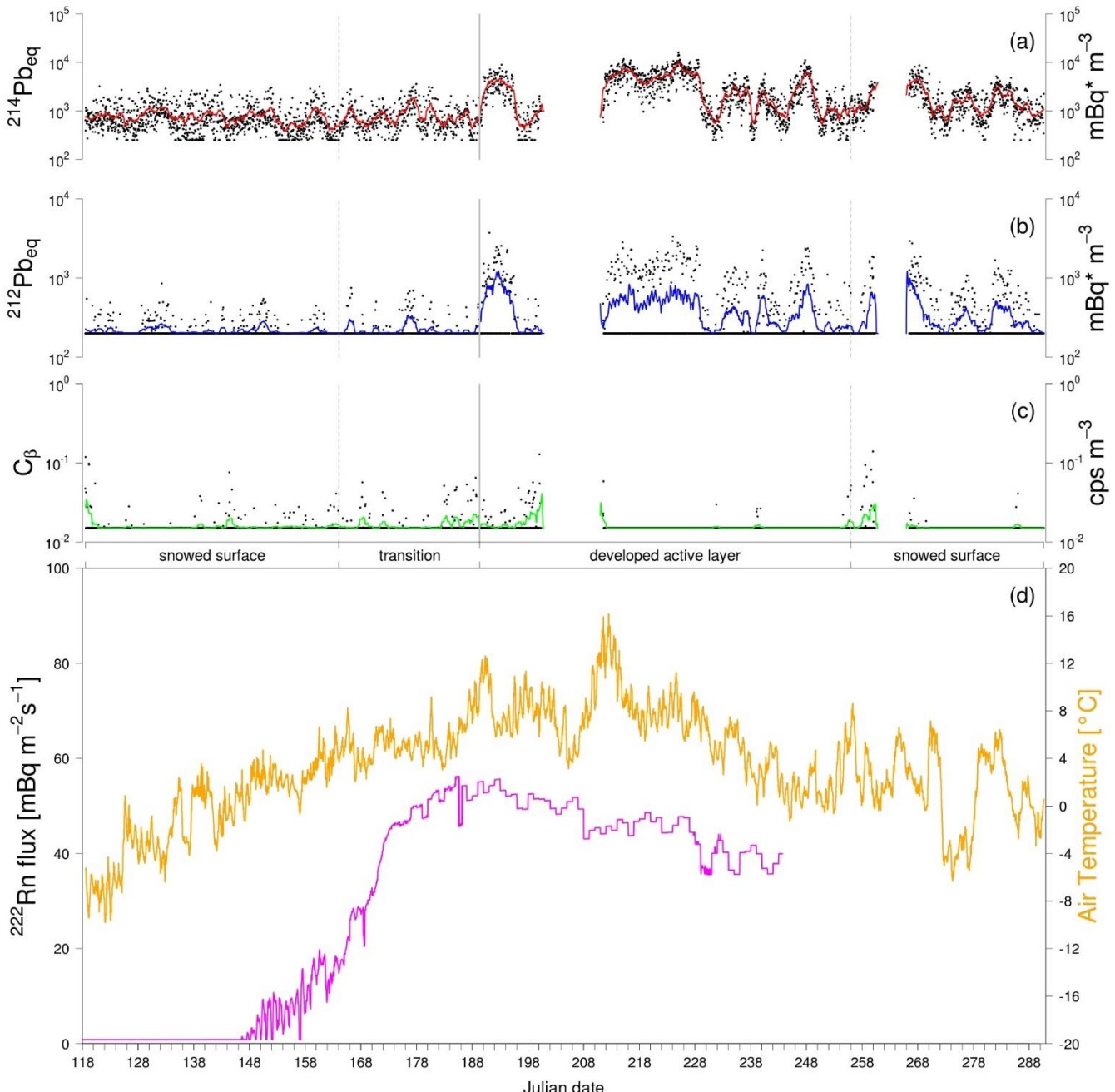

**Figure 2: Variability of the three estimated radioactive components (black points) during the campaign at Ny Ålesund in 2015. Coloured lines show the smoothed trend of $^{214}Pb_{eq}$ (a), $^{212}Pb_{eq}$ (b) and $C_\beta$ (c). Activities are represented assuming equilibrium between progenies and a bold calibration estimation (\*). Comparison (d) with the simulated soil flux (magenta) and the air temperature (orange). Note the log scale for panels a, b and c.**

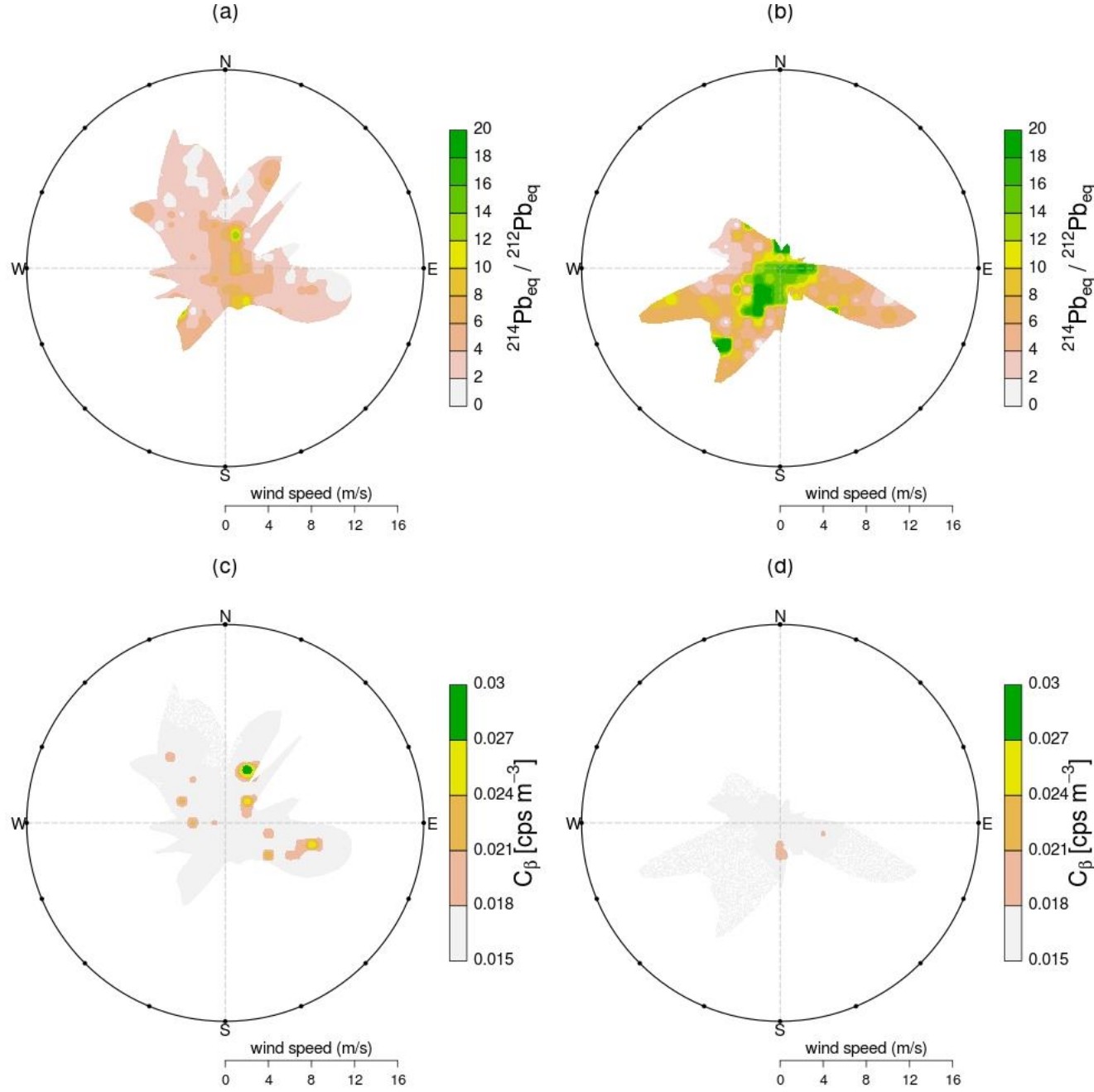

**Figure 3: Polar bivariate plots of $^{214}Pb_{eq}$ - $^{212}Pb_{eq}$ (a, b) and $C_\beta$ (c, d) during the two different emanation periods. The low emanation period (a, c) and the high emanation period (b, d) are based on hourly observations assuming equilibrium between radon progenies.**

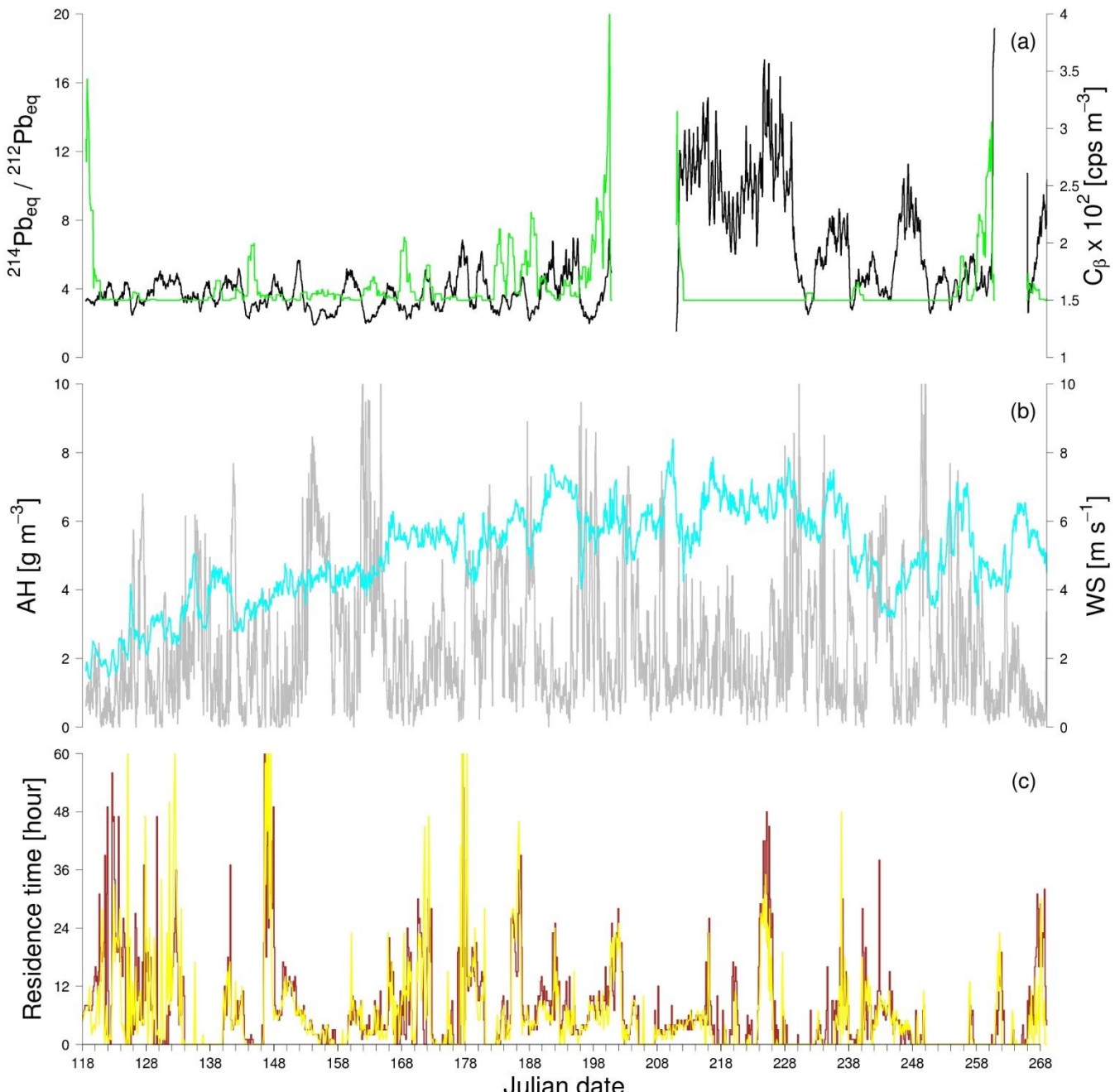

**Figure 4: Evolution of the isotopic composition of radon progeny (a) compared with meteorological parameters (b) and the residence time of air masses close to Svalbard islands (c). The $^{214}Pb_{eq}$ - $^{212}Pb_{eq}$ ratio (black) is reported with the relative counts of $C_\beta$ (green). The wind speed (grey) and the absolute humidity (cyan) are associated with the residence time of air masses over Svalbard (within a distance of 350 km) at different altitudes: 500 m a.s.l. (red) and 1000 m a.s.l. (yellow).**