# Peer review of "High-time resolved radon-progeny measurements in the Arctic region (Svalbard Islands, Norway): results and potentialities"

_Atmospheric Chemistry and Physics, 2017_

## Referee Comment (RC1) · Anonymous Referee #1 · 30 Aug 2017

Journal: ACP Title: High-time resolved radon-progeny measurements in the Arctic region (Svalbard Islands, Norway): results and potentialities Author(s): Roberto Salzano et al. MS No.: acp-2017-668 MS Type: Research article

General comments

The manuscript deals with radon progeny measurements in the High Arctic. The subject is certainly interesting to the readers of ACP and the data obtained during the field work is rare. My main comment is related to the instrument calibration. The results are presented as count rate per cubic meter. The authors should try to convert these to activity concentration units, otherwise comparing the data to other radon progeny

observations is impossible. I understand the difficulties associated with this, detector efficiency for different nuclides, variations in the radon progeny disequilibrium etc. Still, the authors should do this, even with bold assumptions. One way would be comparing the operated instrument to other type but calibrated instruments. This would allow the comparison of activity concentration results to other observations in the Arctic area. Hasn't there been a Heidelberg radon monitor at Mt. Zeppelin monitoring station at Ny-Ålesund?

Spesific comments

I believe the terms NORM and TENORM are usually used with materials associated with human activities, not radionuclides in the atmosphere. An example is oil drilling sludge containing lead-210 or radium-226.

The terms S$\beta$, L$\beta$, and C$\beta$ could be replaced with appropriate IUPAC names after the calibration procedure mentioned above.

Technical comments

In the literature reference list Sthol should be corrected to Stohl.

---

## Author Comment (AC1) · 5 Sep 2017

RC#1: "The authors should try to convert these to activity concentration units, otherwise comparing the data to other radon progeny observations is impossible. I understand the difficulties associated with this, detector efficiency for different nuclides, variations in the radon progeny disequilibrium etc. Still, the authors should do this, even with bold assumptions. One way would be comparing the operated instrument to other type but calibrated instruments. This would allow the comparison of activity concentration results to other observations in the Arctic area."

The calibration of gross-beta counting systems is a critical issue that we are trying to

fix. The logistics is a limiting feature for the definition of a routinely verification of the instrumental calibration. We mentioned in the article about a reference material that supports a preliminary estimation of the detector efficiency, and consequently also the conversion from counts to disintegrations. We preferred to define a more robust and routinely way to estimate efficiency before converting quantities. The paper highlighted at the moment the importance of variations and high-time resolution but further work is required for approaching the disequilibrium issue. This is another problem that avoid comparison between progeny measurements at different sites and conversion to radon and thoron measurements. The best assumption that we can use is the equilibrium between radon progeny ($214Pb = 214Bi$) and thoron progeny ($212Pb = 212Bi$).

RC#2: "Hasn't there been a Heidelberg radon monitor at Mt. Zeppelin monitoring station at Ny-Ålesund?"

We found some projects declaring the activity you are mentioning but no data and publications are available. Could you indicate some contacts?

RC#3: "I believe the terms NORM and TENORM are usually used with materials associated with human activities, not radionuclides in the atmosphere. An example is oil drilling sludge containing lead-210 or radium-226."

The definition of NORM in the IAEA glossary (https://www.iaea.org/ns/tutorials/regcontrol/intro/glossaryn.htm) is "Material containing no significant amounts of radionuclides other than naturally occurring radionuclides. This includes materials in which the activity concentrations of the naturally occurring radionuclides have been changed by man-made processes. These are sometimes referred to as technically enhanced NORM or TENORM and, as a result, the term NORM is sometimes used in contrast with TENORM, i.e. to refer only to materials in which the activity concentrations have not been technologically enhanced". Mining and drilling produce NORM residues and aerosol can be considered a NORM even if human activities are not involved. We can fix this misleading acronym removing NORMs form the text and refer just to naturally occurring radionuclide.

RC#4: "The terms S$\beta$, L$\beta$, and C$\beta$ could be replaced with appropriate IUPAC names after the calibration procedure mentioned above."

The used terms were defined considering half life of possible radionuclides. We could indicate 214Pb* and 212Pb* instead of S$\beta$ and L$\beta$ where * refers to the equivalent activity assuming an equilibrium between the progeny (214Pb = 214Bi and 212Pb = 212Bi)). This cannot be done for C$\beta$ where 210Bi is probably dominant but we need further analyses for supporting such an assumption. It is probably better to be conservative and homogeneous using a definition based on half-life. The IUPAC definition could be invoked when we will have more robust analytical results. We are focused at the moment to the potentialities of high-time resolution.

––––––––––––––––––––––––––––––––––

---

## Short Comment (SC1) · 31 Oct 2017

Review of the discussion paper "High-time resolved radon-progeny measurements in the Arctic region (Svalbard Islands, Norway): results and potentialities" (acp-2017-668) by Salzano et al.

The authors present a 6-month dataset of hourly resolution radon ($222Rn$) and thoron ($220Rn$) progeny measurements from the northwestern part of the Svalbard Islands, and present model results of the influence of permafrost dynamics on the local radon flux. The primary aim of the paper, apart from demonstrating the improved temporal resolution of their radon progeny detector, is to investigate relative contributions of

remote and local terrestrial influence on observed progeny concentrations, with a view to improving understanding of pollutant transport to the polar region through better characterisation of air mass origin and residence time.

I believe that studies of this nature are indeed important, and would be of interest to the readership of Atmospheric Chemistry and Physics. As the paper currently stands, however, I consider that major revision would be required before it is suitable for publication. My main concern is that the investigation of radon progeny concentration variability, given the improved temporal resolution the measurements, is seems cursory and qualitative. Furthermore, the utility of the results presented regarding the broader ACP readership (i.e. those not intimately familiar with radon progeny measurement) will be limited by the fact that no absolute calibration (to activity concentrations) of the observations seems to be available, despite the fact that the type of instrument used in this study (an FAI Instruments PBL mixing monitor) has a well-established history of use in environmental radon progeny detection.

While it may not be possible for the authors to provide calibrated activity concentrations for these measurements, due to some issues they describe with this deployment, in my opinion there is still considerable scope for improving the depth of their investigation of radon progeny concentration variability at this site. In the comments below I make some suggestions regarding how this may be achieved, as well as making some minor corrections where appropriate.

Lastly, at several places in the text (including P2 L25, L30-32) the authors make various claims about the comparative performance or relative utility of the PBL mixing monitor. To assist the authors in more accurately portraying the advantages/disadvantages of their instrument for such applications I have included current performance information regarding two-filter detectors. It is not my intention that the authors include all of this information in their revised text, rather, select from it only what they require in order to make their statements factually correct.

Scott Chambers, Research Scientist Environmental Research, ANSTO

General comments:

Since the novelty of this study lies mostly in the high temporal resolution of the observations, I would expect the authors to make better use of this capability. For example, perhaps diurnal composite plots of the 3 activity concentrations could be prepared for the low and high emission periods, to see what (if any) regular structure is evident, and whether or not this structure can be explained by local diurnal changes in meteorology (which would also require diurnal composite plots of the meteorological components). Perhaps diurnal sampling windows are necessary to help distinguish between local and remote phenomena under some conditions? Speaking of local processes, there is significant topography (order 1000m) adjacent the observation site. Do calculated absolute humidity values and diurnal wind speed/direction indicate the occurrence of katabatic drainage flows at any times of the observation period? If these flows are bringing to the surface air of recent tropospheric origin under certain conditions, is this contributing to the $C_\beta$ observations in any way? The authors allude to orographic effects at the site on P8 L8-9, but make no effort here to investigate the possibility further. Since the $C_\beta$ activities appear so disconnected from the behaviour of the radon progeny, it would be interesting if the authors could say something about what the main driving factors for the observed $C_\beta$ activity actually are at this site.

The authors need to invest more effort to effectively separate local and remote terrestrial influence on their observations (more detailed than the present Fig 3 summary). For example, an hourly ratio between thoron and radon would provide a relative measure of local vs remote influence. Such data could be plotted against wind speed to see whether a wind speed threshold could be used at this site to better separate local and remote influences (after deciding upon a L:S ratio threshold to separate local from remote influences). Local and remote influences could be separately investigated in more detail. For example, a better relationship between simulated local source strengths and observed activities might be obtained if a wind speed threshold was used to isolate the

local signal. Likewise, a more comprehensive (and statistically robust) trajectory analysis (than the present "analysis" that appears to be based on 4 individual trajectories), could be performed on remote terrestrial influences if high $S\beta$ activity periods were targeted within periods of identified remote influence (based on the determined wind speed threshold or L:S ratios).

Plotting the previously mentioned ratios (S:L, S:C, L:C) along with wind speed and direction might also help with a more detailed interpretation of the information summarised in the current Figure 3. Certainly, the "age" of the radon in the sampled air could be effectively demonstrated using the hourly L:S ratio, and periods when the thoron contribution is low (due to a distant influence) could be targeted for separate investigation.

To assist with the authors' intention of further investigating the effects of atmospheric stability on observed activity variability at this site resulting from local contributions, they might consider selecting a defined portion of data (say the period of high radon activity within the first 2 weeks of August), and re-plotting just this portion (so that data features are clearer) along with the corresponding wind speed, temperature and absolute humidity. If there are extended times within this two week period when wind speeds are ≤3 m s-1, then the authors might consider approximating and removing fetch effects as described in Chambers et al. (2015), and investigating the resultant diurnal variability of radon activity for radon accumulation periods. They may have some success in relating these radon accumulation periods to their predicted fluxes (if estimates of mixing depth can be made).

Lastly, overall the article would benefit greatly from a proof reading by a native English speaker to improve the grammar and flow.

Specific comments:

P1 L10-11: The authors draw attention to the stringent requirements of radon lower limit of detection for measurements in the Arctic. Briefly in the Introduction, for context, the

authors might like to quantify what they believe to be the required LLD for observations of this kind in the Arctic, how this differs from LLD requirements in the Antarctic, and why this is the case (making reference to the potential range / restrictions of possible terrestrial-free fetch; since this is pertinent to their general interest in pollution transport to Arctic regions).

Radon concentration thresholds for "baseline" (minimally terrestrially influenced) or "regional background" air masses are becoming more clearly established (see, for example, Chambers et al. 2016a and references therein). Since calibrated activity concentrations are not provided in this study it makes it harder for the reader to estimate, relative to other studies, the degree of recent (within the past 2 weeks) terrestrial influence from unfrozen surfaces the observed air masses have experienced. Can the authors help to bridge this gap by approximating what range of radon concentrations their observed radon activity values in Fig 2 represent?

P1 L12: A claim to uniqueness of this study is the ability to resolve, at hourly temporal resolution, the activities of different radon progeny ($220Rn$, $222Rn$) at concentrations typical of the Arctic. But aren't there other readily available single-filter radon progeny detectors that capable of doing the same? One example that comes to mind is the Heidelberg Radon Monitor (HRM; Levin et al. 2002); the output of which can be readily calibrated to radon progeny activity concentrations. HRM's have been successfully deployed and operated at several Antarctic bases (for which LLD requirements are more stringent than in the Arctic). If the FAI Instruments PBL mixing monitor (in the configuration adopted for this study), has capabilities significantly beyond those of other such monitors, it would indeed be worthwhile for the authors to make this point clearly. Furthermore, direct electrostatic deposition monitors (e.g. Wada et al. 2010; Grossi et al. 2012) are also capable of separately resolving these radon isotopes, are relatively portable, require no assumptions about the degree of equilibrium between radon and its progeny, and have a lower limit of detection comparable to the FAI PBL mixing monitor. Does the PBL mixing monitor have any particular advantages over these

kinds of detectors? (I ask this question in relation to the quote from the authors that I have copied below in my comment on "P2 L30-32")

P1 L 28-29: Some other articles pertaining to the application of radon observations in atmospheric stability analyses that may be of interest to the authors include Williams et al. (2016), Wang et al (2016), and Chambers et al. (2016b,c).

P2 L 4: Regarding detectors capable of very low level radon detection for polar or high-altitude environments, and their applications, the authors can find further, more up to date, information in Williams & Chambers (2016); Chambers et al. (2016a).

P2 L6: Regarding direct detection methods. The direct ANSTO dual-flow-loop two-filter radon detectors actually observe the alpha decay of both the 218Po (t0.5 ~3 min) and 214Po (t0.5 ~20 min) progeny of 222Rn (see Griffiths et al. 2016 for details). However, since they are incapable of distinguishing between alpha particles of different energy, thoron (220Rn) is removed from the sample air prior to entering the detector. Detector response time issues related to the half-lives of the two radon progeny mentioned above can be completely corrected for as described in Griffiths et al. (2016). Importantly, direct techniques generally observe radon progeny formed under controlled (aerosol-free) conditions within their measurement delay volumes where radon gas is in equilibrium with its unattached progeny.

P2 L9: Since radon is a noble gas, presumably it is the physical rather than chemical behaviour of radon upon which these techniques rely?

P2 L12: The way the parentheses are placed here makes it seem like radon and thoron are their own decay products.

P2 L19: Reference missing for the citation of Wada et al. (2010). Please check all references.

P2 L19: As described in Williams and Chambers (2016) the lowest detection limit for continuous, high temporal resolution, environmental atmospheric radon concentration

measurements is actually less than 10 mBq m-3; not 70 mBq m-3 as quoted by the authors. However, the 5000 L detector capable of such observations is strictly one of a kind, and operates only at the Cape Grim Baseline Air Pollution Station. The lowest detection limit for a routinely available ANSTO dual-flow-loop two-filter radon detector (the 1500 L model) is around 25 mBq m-3 (see, for example, Chambers et al. 2014; 2016a). When response time corrected (as per Griffiths et al. 2016) these detectors have a temporal resolution of 30 minutes and an absolute accuracy of around 10% at radon concentrations of 100 Bq m-3 (as described in Chambers et al. (2014) this accuracy further improves for longer averaging times or higher concentrations).

P2 L22: Please note that the terms $S_\beta$, $L_\beta$ and $C_\beta$ have not yet been defined in the manuscript.

P2 L23: I feel that this brief review of radon detection technology is incomplete without mention of the Heidelberg Radon Monitor (Levin et al. 2002; see also Schmithüsen et al. (2017) for a discussion of many of the research-grade radon detectors currently operating throughout Europe; details of the ARMON electrostatic deposition detectors operating in Spain are available in Grossi et al. 2012).

P2 L25: "... the lowest detection limits can [only] be obtained having a complex sampling/counting system that is difficult to deploy and maintain in remote conditions"

I believe that this statement is incorrect.

The only disadvantages of two-filter detectors (capable of the lowest detection limits) are (i) that they are not readily portable (after having been installed), on account of their large size (2-3m), and (ii) and that they measure only Radon-222 (since Radon-220 is removed from the sampled air stream prior to entering the detector). 220Rn removal is necessary because their alpha counting system can't distinguish between ïĄą-particles of different energy.

The operation of the two-filter detectors is not complex; it is based primarily on a ZnS-

photomultiplier counting system, a pair of centrifugal blowers, and a Campbell data logger. As such, power requirements are limited to around 100-120W at 240V when sampling from close to the surface. In spite of their size, these detectors weigh only around 100 kg, and can be readily deployed in challenging remote sites (from mountain-top to polar regions) or mobile platforms (such as ships). Furthermore, where network services are available they can be fully remotely controlled. Since calibration and instrumental background checks on the two-filter detectors are performed automatically (or via remote control), maintenance requirements are also minimal. In fact, a 1500 L model two-filter radon detector has been in service in Antarctica since February 2013 to current (October 2017), and the only user intervention required over this >4-year period has been to remove ice collected on the inlet tube on two occasions. Over this time the detector's calibration has remained quite stable, as has the lower limit of detection (25-30 mBq m-3). In most situations, however, we have found it prudent to replace the sensitive components of the two-filter detector's measurement head every 5 years to maintain a high sensitivity and low instrumental background.

P2 L28-29: Particular assumptions regarding the degree of equilibrium between radon and its progeny will also change under high humidity (or indeed foggy or hazy) conditions, and (during the summer months at this site when local emissions are significant), depending on the height above ground at which sampling is conducted.

P2 L30-32: "This is a single-filter approach coupled to beta-counting and it represents, at the moment, the best compromise between detection efficiency and required resources."

This claim, I feel, is somewhat misleading.

As previously mentioned, two-filter detectors have low power requirements, minimal maintenance requirements, a 30 minute temporal resolution, require no assumptions to be made about the state of equilibrium between radon and its progeny, an average measurement sensitivity that rarely changes by more than 1% per year (in a roughly

linear, correctable manner), and have a detection limit almost an order of magnitude better than that of the FAI PBL mixing monitor. They are, however, large (if space is an issue at the measurement location), not designed to be portable (which is only really a concern for short-term campaigns, since unpacking and initial setup can take 2 days), and they are not capable of monitoring activity concentrations of thoron progeny, or cosmogenic radionuclides. In summary, there are some advantages to using the PBL mixing monitor rather than a two-filter detector in some situations, but I think these relate more to its portability and ability to distinguish between different progeny than to resource requirements (e.g. maintenance and power).

Interestingly, in their comparison of advantages/disadvantages between direct and indirect measurements, the authors fail to mention the apparent difficulty in obtaining consistent absolute radon activity concentrations from the instrument used in this study. Following claims that the instrument is readily deployable in remote environments, and that it requires minimal maintenance/resources, later (on page 5) the authors go on to say "Considering the logistic restrictions of the study site, routine quality check and sampling efficiency assessments were not possible." Problems, apparently specific to this campaign, that have prevented the authors from reporting of absolute radon concentrations in this study. However, despite the established history in the literature of applying the FAI PBL mixing monitor for atmospheric radon sampling (and other similar single-filter $\beta$-radiation detectors of this kind, such as the OPSIS SM200 stability monitor), few of the published studies report calibrated (absolute) radon activity concentrations. It would certainly improve the utility of these devices for applications like the one described in this study if absolute calibration of the observations was routinely possible.

P3 L7: Regarding Figure 1b, this figure would be more useful to the reader if the view were "zoomed out" a little more. If the figure was changed such that the width represented 150-200 km, instead of about 50 km, then it would put the site in better context regarding the trajectory analysis and local influences, and would not lose too

much of the local topographic detail.

Section 2.1: since this study is not the first application of the FAI PBL monitor, please include only the detail and theory in this section that (i) has not already been published, and (ii) pertains to the unique features of the detector operation for this study (which, as I understand, is the increased temporal resolution of sampling). Perhaps all of the detail in this section is required (the authors would be the best judge), but if other publications summarise the theory of operation (as much as it is similar to the FAI PBL mixing monitors with the slower temporal response), then it would be sufficient to refer the reader to other published works for an overview of the theory or principle of operation. This may leave more room for a more detailed analysis of the observations later.

P6 L10-12: Can the authors provide any indication of how "good" the remote soil moisture estimates are? Was there any ground-truthing performed (either for this study or in the literature)? A reference to a study where the technique has been evaluated would be sufficient if nothing specific was tested in this study.

P6 L16: Could the authors comment briefly on the results of the comparisons of trajectory calculations between 500 and 1000m that led them to their final choice?

P6 L22-23: "The evolution of the three radioactive components (Fig. 2a) seemed to be produced by the overlapping of different sources and processes."

This may well be the case, but little evidence to support this statement is provided in Figure 2a. Modelled local radon flux and air temperature are provided as companion series to the activity measurements, but there appears to be little in the way of direct consistent correlations between either of these two parameters and the more significant of the reported concentration variations in the measured activities. Perhaps including time series of wind speed, wind direction, ratios (e.g. between S:L, S:C, L:C), or trajectory-modelled time-over-land for each sample over the past 5 days would provide more information about factors contributing to the observed variability?

[Figure]

Regarding Figure 2, please rethink the scale of the x-axis, consider decimal days or something similar. There appears to be little relationship between the axis tick marks and labels. This makes it hard to relate them to the data.

P6 L24-28: Various analyses are mentioned here, but there is no evidence of them in the figures (i.e. before/after plots showing the effect of what has been achieved, and why it was necessary).

References

Chambers, SD, Hong, S-B, Williams, AG, Crawford, J, Griffiths, AD, and Park, S-J.: 'Characterising terrestrial influences on Antarctic air masses using Radon-222 measurements at King George Island', Atmos. Chem. Phys., 14, 9903-9916, 2014.

Chambers, SD, Williams, A. G., Crawford, J., and Griffiths, A. D.: 'On the use of radon for quantifying the effects of atmospheric stability on urban emissions', Atmos. Chem. Phys., 15, 1175–1190, 2015.

Chambers, SD, Williams, AG, Conen, F, Griffiths, AD, Reimann, S, Steinbacher, M, Krummel, PB, Steele, LP, van der Schoot, MV, Galbally, IE, Molloy, SB, and Barnes, JE. 'Towards a universal "baseline" characterisation of air masses for high- and low-altitude observing stations using Radon-222'. Aerosol and Air Quality Research 16, 885–899, 2016a.

Chambers SD, Podstawczynska A, Williams AG and Pawlak W. 'Characterising the influence of atmospheric mixing state on Urban Heat Island Intensity using Radon-222', Atmos Env 147, 355-368, 2016b.

Chambers, SD, Galeriu, D, Williams, AG, Melintescu, A, Griffiths, AD, Crawford, J, Dyer, L, Duma, M, and Zorila, B: 'Atmospheric stability effects on potential radiological releases at a nuclear research facility in Romania: characterising the atmospheric mixing state', Journal of Environmental Radioactivity, 154, 68-82, 2016c.

Griffiths AD, Chambers SD, Williams AG and Werczynski SR. 'Increasing the accuracy

and temporal resolution of two-filter radon–222 measurements by correcting for the instrument response', Atmos. Meas. Tech. 9, 2689-2707, 2016.

Grossi, C., Arnold, D., Adame, A. J., Lopez-Coto, I., Bolivar, J. P., de la Morena, B. A., and Vargas, A.: Atmospheric 222Rn concentration and source term at El Arenosillo 100m meteorological tower in southwest, Spain. Radiat. Meas., 47, 149–162, doi:10.1016/j.radmeas.2011.11.006, 2012.

Levin, I., Born, M., Cuntz, M., Langendörfer, U., Mantsch, S., Naegler, T., Schmidt, M., Varlagin, A., Verclas, S. and Wagenbach, D. "Observations of atmospheric variability and soil exhalation rate of Radon-222 at a Russian forest site: Technical approach and deployment for boundary layer studies". Tellus 54B, 462-475, 2002.

Schmithüsen D, Chambers SD, Fischer B, Gilge S, Hatakka J, Kazan V, Neubert R, Paatero J, Ramonet M, Schlosser C, Schmid S, Vermeulen A, and Levin I, 2016: 'A European-wide 222Rn and 222Rn progeny comparison study', Atmospheric Measurement Techniques, 10, 1299-1312, 2017.

Wada, A., Muramaya, S., Kondo, H., Matsueda, H., Sawa, Y., Tsuboi, K., 2010. Development of a compact and sensitive electrostatic radon-222 measuring system for use in atmospheric observation. J. Meteorol. Soc. Jpn. 88 (2), 123e134. http://dx.doi.org/10.2151/jmsj.2010-202.

Wang F, Chambers SD, Zhang Z, Williams AG, Deng X, Zhang H, Lonati G, Crawford J, Griffiths AD, Ianniello A, and Allegrini I. 'Quantifying stability influences on air pollution in Lanzhou, China, using a radon-based "stability monitor": Seasonality and extreme events', Atmospheric Environment 145, 376-391, 2016.

Williams, AG, and Chambers, SD. 'A history of radon measurements at Cape Grim', Baseline Atmospheric Program (Australia) History and Recollections (40th Anniversary Special Edition), 131-146, 2016.

Williams AG, Chambers SD, Conen F, Reimann S, Hill M, Griffiths AD, and Crawford J.

'Radon as a tracer of atmospheric influences on traffic-related air pollution in a small inland city', Tellus B 68, 30967, 2016.

---

## Author Comment (AC2) · 16 Nov 2017

The aim of this study is to highlight the potentialities of high-time resolved measurements of radon progeny in the Arctic region. Furthermore, our intention is to outline the importance of radon-derived information in remote areas where this kind of information are still limited. We know that we considered a single-filter technique that has a lot limitations concerning detection limits and absolute estimation of radon concentration. From this perspective, we cannot support that single-filter methods are better than dual-filter systems that represent probably (for the reasons mentioned in the comments) the best available technologies. The term "best compromise" was referred

to our logistical resources and we will use all your suggestion to better focus advantages/disadvantages of our technique. Concerning the results, the novelty of our approach, respect to the 20-years literature about single-filter measurements, consists in the introduction of the near-constant decay component. All the presented calculations are, in fact, not included in FAI instruments or similar systems. From this perspective, we think that there is a significant contribution for investigating variations in terms of natural radioactivity in areas where the Arctic haze occurs. Some of your comments will surely help us to increase the depth of our data analysis. Thank you in advance.

---

## Referee Comment (RC2) · Anonymous Referee #2 · 25 Jan 2018

Define Sbeta, Lbeta and Cbeta at the first appearance. Actually they are defined much later, above Eq. 5. Do not use "radon daughter". Instead should be used "Radon progeny". Please give more details about Eqs. 6. How were these equations derived. From methodology section is not clear whether particular radon/thoron progeny determined, (i.e. could you determine Po218, Pb214 Bl214 etc) or you determine just total sum of beta counts due to radon, thoron and cosmogenic nuclide. Above Eq 1 was written that Tbeta is number of beta particles emitted by different nuclides. However, later in Eq 7, TïĄ́c has somewhat different meaning. In Eq. 7 TïĄ́c is number of counts due beta emitters in first and fourth measuring intervals. In Eq. 7 progeny concentration was multiplied with detection efficiency which produce count numbers.

What is the sense of decay parameters ($\delta\dot{I}\acute{S}\acute{S}\delta\dot{I}\acute{S}\H{U}\delta\dot{I}\acute{S}\.{Z}$) in Eq. 7 is not clear- please explain. I have experience with radon progeny measurements from beta emitters on filter. Very often, some physically non realistic results were obtained- due to i) variation of detection efficiency because of beta spectrum changing during the counting, and ii) counting statistic which is important source of errors particularly when the count rate is small. Authors devote significant care to the variation of detection efficiency, but the second fact is unavoidable. I can assume that count rates in measuring intervals are small, due to small radon concentration in open space. Then, statistical variations are large while this method is very sensitive on the number of counts. I would like to know did authors meet some physically unacceptable results or not. Bellow Eq. 9 was written "The estimation of the three components was obtained minimizing the chi squared indicator, calculated between the four counting intervals and the respective values simulated between the two endmember situations ". Can you explain in more details what is the meaning of the previous sentence. From this sentence follows that values were simulated. Then what was the purposes of the measurements. Counts obtained in 2nd and 3rd intervals were not used in calculations. Does this mean that those counts were taken from simulation (not from measurements) in order to avoid physically non realistic results.

I am not expert in climatology. so I will not comment second part of ms, which is related to trajectory of air masses etc.

---

## Referee Comment (RC3) · Chambers (Referee) · 29 Jan 2018

The comment was uploaded in the form of a supplement: https://www.atmos-chem-phys-discuss.net/acp-2017-668/acp-2017-668-RC3-supplement.pdf

---

## Author Response (AR1)

**Reviewer #1**

RC1.1: "The manuscript deals with radon progeny measurements in the High Arctic. The subject is certainly interesting to the readers of ACP and the data obtained during the field work is rare. My main comment is related to the instrument calibration. The results are presented as count rate per cubic meter. The authors should try to convert these to activity concentration units, otherwise comparing the data to other radon progeny observations is impossible. I understand the difficulties associated with this, detector efficiency for different nuclides, variations in the radon progeny disequilibrium etc. Still, the authors should do this, even with bold assumptions. One way would be comparing the operated instrument to other type but calibrated instruments. This would allow the comparison of activity concentration results to other observations in the Arctic area."

We know that activity conversion is a bold limitation, but we are interested on variations not only on absolute determinations. We cannot move easily radioactive materials, useful for the calibration procedure, to Svalbard island and we are not in contact with groups with different instruments that can support interesting intercomparison with our setup. We know that conversion is important and for this reason we expressed activities in mBq m-3 even with the bold assumptions expressed in a note (\*).

RC1.2: "Hasn't there been a Heidelberg radon monitor at Mt. Zeppelin monitoring station at Ny-Ålesund?"

We found some projects (ARCTOC and RIS-1035) declaring the activity you are mentioning but no data and publications are easily accessible. We found the project report where radon is included but any result is described.

RC1.3: "I believe the terms NORM and TENORM are usually used with materials associated with human activities, not radionuclides in the atmosphere. An example is oil drilling sludge containing lead-210 or radium-226."

We removed NORMs form the text and referred only to "naturally-occurring radionuclides".

RC1.4: "The terms S $\beta$ , L $\beta$ , and C $\beta$  could be replaced with appropriate IUPAC names after the calibration procedure mentioned above."

The used terms were defined considering half-life of possible radionuclides. We expressed S $\beta$  and L $\beta$  in terms of 214Pbeq and 212Pbeq adding an "eq" suffix in order to distinguish our results stating the bold assumption of equilibrium between nuclides.

RC1.5: "In the literature reference list Sthol should be corrected to Stohl."

Done

**Reviewer #2**

RC2.1: "Define Sbeta, Lbeta and Cbeta at the first appearance. Actually they are defined much later, above Eq. 5."

**Done in page 2**

RC2.2: "Do not use "radon daughter". Instead should be used "Radon progeny"."

**We fixed this indication**

RC2.3: "Please give more details about Eqs. 6. How were these equations derived. From methodology section is not clear whether particular radon/thoron progeny determined, (i.e. could you determine Po218, Pb214 BI214 etc) or you determine just total sum of beta counts due to radon, thoron and cosmogenic nuclide."

Equations from 1 to 7 are theoretical formulations that explain how to obtain the operational equations 8a,b,c. We have gross beta counts and we try to separate three different contributions depending on the specific half-life of nuclides. We cannot determine directly the separated activites of each isotope.

RC2.4: "Above Eq 1 was written that Tbeta is number of beta particles emitted by different nuclides. However, later in Eq 7, Tbeta has somewhat different meaning. In Eq. 7 Tbeta is number of counts due beta emitters in first and fourth measuring intervals."

We checked definitions in order to be coherent. Tbeta in Eq.1 is the sum of beta particles generated by different nuclides (see text in P3 L3). It is the sum of three defined radioactive components Sbeta, Lbeta and Cbeta above Eq.5 and if we consider different counting intervals (the first and the forth) we have to specify Tbeta as mentioned in the text for Eq 7.

RC2.5: "In Eq. 7 progeny concentration was multiplied with detection efficiency which produce count numbers. What is the sense of decay parameters (ðiŚŚðiŚŰðiŚŻ) in Eq. 7 is not clear, please explain. I have experience with radon progeny measurements from beta emitters on filter. Very often, some physically non realistic results were obtained- due to i) variation of detection efficiency because of beta spectrum changing during the counting, and ii) counting statistic which is important source of errors particularly when the count rate is small. Authors devote significant care to the variation of detection efficiency, but the second fact is unavoidable. I can assume that count rates in measuring intervals are small, due to small radon concentration in open space. Then, statistical variations are large while this method is very sensitive on the number of counts. I would like to know did authors meet some physically unacceptable results or not."

The decay parameter is a coefficient derived for each counting interval considering the sampling phase (Eq.2) and counting phase (Eq.4). We specified this relation in the text. The counting statistics is of course important when activities are low and this issue represents a limitation of this technique especially regarding the long-lived progeny and the near-constant progeny. The background levels of those components are probably close to the specific LLDs and an high-frequency bias can occur due to this limitation.

RC2.5: "Bellow Eq. 9 was written "The estimation of the three components was obtained minimizing the chi squared indicator, calculated between the four counting intervals and the respective values simulated between the two endmember situations". Can you explain in more details what is the meaning of the previous sentence. From this sentence follows that values were

simulated. Then what was the purposes of the measurements. Counts obtained in 2nd and 3rd intervals were not used in calculations. Does this mean that those counts were taken from simulation (not from measurements) in order to avoid physically non realistic results."

We clarified in the text that while the endmembers situation were defined considering only the 1st and 4th counting intervals, the final estimation of the three components were selected considering the RMSE calculated using all of the four observed counting intervals. This procedure supported the minimization of biases associated with low counting levels. When counting levels were low we still have an high frequency bias that can be removed only having a solid calibration procedure and possibly an intercomparison with an independent technique.

**Reviewer #3**

RC3.1: "For example, perhaps diurnal composite plots of the 3 activity concentrations could be prepared for the low and high emission periods, to see what (if any) regular structure is evident, and whether or not this structure can be explained by local diurnal changes in meteorology (which would also require diurnal composite plots of the meteorological components). Perhaps diurnal sampling windows are necessary to help distinguish between local and remote phenomena under some conditions?"

We considered short-lived activities above the 90th percentile, the most stable atmospheric conditions, and any diurnal pattern is evident during the two emanation periods, coherently with the length of the daylight (about 24 hours) during the considered period. Except for air temperature, meteorological parameters (absolute humidity and wind speed) and activities are almost stationary in the composite plot. There are of course high-frequency turbulences but making averages, this information is lost. More detailed information must be acquired in order to investigate diurnal patterns but the complexity of the fiord system dominates on the diurnal variations.

RC3.2: "Speaking of local processes, there is significant topography (order 1000m) adjacent the observation site. Do calculated absolute humidity values and diurnal wind speed/direction indicate the occurrence of katabatic drainage flows at any times of the observation period?"

Topography of Svalbard, focusing the attention specifically to the study site, is characterized by the maximum elevation of about 1000 m a.s.l. and the presence of small glaciers (below 100km2) close to the observatory. The meteorological description of the site [Beine et al 2001; Argentini et al 2001] associates the dominant source of katabatic events with the Kongsvegen (105km2) / Kronebreen (690 km2) glaciers, located to the eastern sector of our site, 15 km far from our facility. There is of course a WNW wind component related to the open sea and secondary slope stream from the SW direction associated with the Broeggerbreen (6 km2) and the Blomstrandbreen (18 km2) glaciers. The katabatic events are more frequently during the winter season and they are almost absent during the summer. A strong channelled behaviour of the wind pattern can be observed due to the morphology of the studied fiord but the penetration of katabatic winds below the inversion layer (ranging between 200 and 500 m a.s.l.) is not so frequent. Esau and Repina (2012) observed a decoupling of wind patterns at the ground and above the inversion layer and associate this behaviour also to the presence of sea ice in the fiord. We added to figure 4 also the panel showing the wind direction and the absolute humidity and commented the observations in section 3.1.

RC3.3: "If these flows are bringing to the surface air of recent tropospheric origin under certain conditions, is this contributing to the C $\beta$  observations in any way? The authors allude to orographic effects at the site on P8 L8-9, but make no effort here to investigate the possibility further."

We analysed the relation between radioactive components and meteorological parameters and high- $C\beta$  events started when significant absolute humidity reductions and wind speed increments occurred. Furthermore, the study site represents a complex system as described in the text and the dominant direction of origin concerning high- $C\beta$  transport is ESE.

RC3.4: "Since the C $\beta$  activities appear so disconnected from the behaviour of the radon progeny, it would be interesting if the authors could say something about what the main driving factors for the observed C $\beta$  activity actually are at this site."

We observed events that can be related to glacier streams (see R3.3) and we can also suppose a contribution of the inversion layer to the income of C $\beta$ -enriched air masses but further information must be gathered in order to fully analyse this contribution to the fiord system.

RC3.5: "The authors need to invest more effort to effectively separate local and remote terrestrial influence on their observations (more detailed than the present Fig 3 summary). For example, an hourly ratio between thoron and radon would provide a relative measure of local vs remote influence. Such data could be plotted against wind speed to see whether a wind speed threshold could be used at this site to better separate local and remote influences (after deciding upon a L:S ratio threshold to separate local from remote influences)."

Two separate thresholds can be defined for each emanation period. This indication is connected also to the wind direction and for this reason we substituted Fig.3 with bivariate polar plots containing the S:L ratio.

RC3.6: "Local and remote influences could be separately investigated in more detail. For example, a better relationship between simulated local source strengths and observed activities might be obtained if a wind speed threshold was used to isolate the local signal. Likewise, a more comprehensive (and statistically robust) trajectory analysis (than the present "analysis" that appears to be based on 4 individual trajectories), could be performed on remote terrestrial influences if high S $\beta$  activity periods were targeted within periods of identified remote influence (based on the determined wind speed threshold or L:S ratios)."

The definition of a wind threshold is controlled also by the wind direction (new Fig.3) since wind coming from the open sea (NW) are less influenced by local contributions compared with air masses coming from the glacier (ESE) that can flow over land much more than oceanic masses. This last origin can involve or not the inversion layer and additional information, absolute humidity do not completely describe this dynamics, are required.

RC3.7: "Plotting the previously mentioned ratios (S:L, S:C, L:C) along with wind speed and direction might also help with a more detailed interpretation of the information summarised in the current Figure 3. Certainly, the "age" of the radon in the sampled air could be effectively demonstrated using the hourly L:S ratio, and periods when the thoron contribution is low (due to a distant influence) could be targeted for separate investigation."

We described the new Fig.3 in the text.

RC3.8: "To assist with the authors' intention of further investigating the effects of atmospheric stability on observed activity variability at this site resulting from local contributions, they might consider selecting a defined portion of data (say the period of high radon activity within the first 2 weeks of August), and re-plotting just this portion (so that data features are clearer) along with the corresponding wind speed, temperature and absolute humidity. If there are extended times within this two week period when wind speeds are  $\leq 3 \text{ m s-1}$ , then the authors might consider approximating and removing fetch effects as described in Chambers et al. (2015), and investigating the resultant diurnal variability of radon activity for radon accumulation periods. They may have some success in relating these radon accumulation periods to their predicted fluxes (if estimates of mixing depth can be made)."

The suggestion is interesting but an adaptation of Chambers et al (2015) is required since diurnal patterns are very limited. We would focus the attention on accumulation periods (the duration can cover up to 20 days during summer) in the next future.

RC3.9: "P1 L10-11: The authors draw attention to the stringent requirements of radon lower limit of detection for measurements in the Arctic. Briefly in the Introduction, for context, the authors might like to quantify what they believe to be the required LLD for observations of this kind in the Arctic,

how this differs from LLD requirements in the Antarctic, and why this is the case (making reference to the potential range / restrictions of possible terrestrial-free fetch; since this is pertinent to their general interest in pollution transport to Arctic regions).

We added to P1 the text in L5-10. added ref Samuelson et al 1986

RC3.10: "Radon concentration thresholds for "baseline" (minimally terrestrially influenced) or "regional background" air masses are becoming more clearly established (see, for example, Chambers et al. 2016a and references therein). Since calibrated activity concentrations are not provided in this study it makes it harder for the reader to estimate, relative to other studies, the degree of recent (within the past 2 weeks) terrestrial influence from unfrozen surfaces the observed air masses have experienced. Can the authors help to bridge this gap by approximating what range of radon concentrations their observed radon activity values in Fig 2 represent?"

**We updated Fig.2**

RC3.11: "P1 L12: A claim to uniqueness of this study is the ability to resolve, at hourly temporal resolution, the activities of different radon progeny (220Rn, 222Rn) at concentrations typical of the Arctic. But aren't there other readily available single-filter radon progeny detectors that capable of doing the same? One example that comes to mind is the Heidelberg Radon Monitor (HRM; Levin et al. 2002); the output of which can be readily calibrated to radon progeny activity concentrations. HRM's have been successfully deployed and operated at several Antarctic bases (for which LLD requirements are more stringent than in the Arctic). If the FAI Instruments PBL mixing monitor (in the configuration adopted for this study), has capabilities significantly beyond those of other such monitors, it would indeed be worthwhile for the authors to make this point clearly. Furthermore, direct electrostatic deposition monitors (e.g. Wada et al. 2010; Grossi et al. 2012) are also capable of separately resolving these radon isotopes, are relatively portable, require no assumptions about the degree of equilibrium between radon and its progeny, and have a lower limit of detection comparable to the FAI PBL mixing monitor. Does the PBL mixing monitor have any particular advantages over these kinds of detectors? (I ask this question in relation to the quote from the authors that I have copied below in my comment on "P2 L30-32")

We specified in the text that our measurements are the first in the Arctic region, defined by latitudes above 70°N. We found the ARCTOC projects and the relative activity in the RiS portal (RIS-1035) declaring attempts to measure radon in Svalbard but no data are easily accessible (no results are present also in the project report). The northernmost dataset is located in Pallas (Finalnd) but latitude is 68°N (Schmithüsen et al. 2017) and the location is a continental site. Levin et al 2002 made observations below 60°N.

RC3.12: "P1 L 28-29: Some other articles pertaining to the application of radon observations in atmospheric stability analyses that may be of interest to the authors include Williams et al. (2016), Wang et al (2016), and Chambers et al. (2016b,c)."

**We added refs of interest.**

RC3.13: "P2 L 4: Regarding detectors capable of very low level radon detection for polar or highaltitude environments, and their applications, the authors can find further, more up to date, information in Williams & Chambers (2016); Chambers et al. (2016a)."

we added refs

RC3.14: "P2 L6: Regarding direct detection methods. The direct ANSTO dual-flow-loop two-filter radon detectors actually observe the alpha decay of both the 218Po (t0.5  $\sim$ 3 min) and 214Po (t0.5  $\sim$ 20 min) progeny of 222Rn (see Griffiths et al. 2016 for details). However, since they are incapable of distinguishing between alpha particles of different energy, thoron (220Rn) is removed from the sample air prior to entering the detector. Detector response time issues related to the half-lives of the two radon progeny mentioned above can be completely corrected for as described in Griffiths et al. (2016). Importantly, direct techniques generally observe radon progeny formed under controlled (aerosol-free) conditions within their measurement delay volumes where radon gas is in equilibrium with its unattached progeny."

We added 214Po to the text and we specified the direct methods foresees a delay volume for obtaining controlled aerosol-free conditions during the measurement.

RC3.15: "P2 L9: Since radon is a noble gas, presumably it is the physical rather than chemical behaviour of radon upon which these techniques rely?"

We changed "chemical" to "physico-chemical" behavior of radon and specified later the radon is "physically-fixed" on aerosols.

RC3.16: "P2 L12: The way the parentheses are placed here makes it seem like radon and thoron are their own decay products."

We anticipated the description of radon decay-half lives to P1 L26 in order to avoid misunderstandings.

RC3.17: "P2 L19: Reference missing for the citation of Wada et al. (2010). Please check all references."

**We added this ref**

RC3.18: "P2 L19: As described in Williams and Chambers (2016) the lowest detection limit for continuous, high temporal resolution, environmental atmospheric radon concentration measurements is actually less than 10 mBq m-3; not 70 mBq m-3 as quoted by the authors. However, the 5000 L detector capable of such observations is strictly one of a kind, and operates only at the Cape Grim Baseline Air Pollution Station. The lowest detection limit for a routinely available ANSTO dual-flow-loop two-filter radon detector (the 1500 L model) is around 25 mBq m-3 (see, for example, Chambers et al. 2014; 2016a). When response time corrected (as per Griffiths et al. 2016) these detectors have a temporal resolution of 30 minutes and an absolute accuracy of around 10% at radon concentrations of 100 Bq m-3 (as described in Chambers et al. (2014) this accuracy further improves for longer averaging times or higher concentrations)."

We updated the LLD with the one declared in Williams and Chambers (2016) referencing to Griffith et al (2016) that is more easily accesssible.

RC3.19: "P2 L22: Please note that the terms S $\beta$ , L $\beta$  and C $\beta$  have not yet been defined in the manuscript."

We substituted the not-declared terms with short-lived, long-lived and near-constant progenies.

RC3.20: "P2 L23: I feel that this brief review of radon detection technology is incomplete without mention of the Heidelberg Radon Monitor (Levin et al. 2002; see also Schmithüsen et al. (2017) for a discussion of many of the research-grade radon detectors currently operating throughout Europe;

details of the ARMON electrostatic deposition detectors operating in Spain are available in Grossi et al. 2012)."

**We added Levin et al (2002) and also Paatero et al (1994) to the review**

RC3.21: "P2 L25: ". . . the lowest detection limits can [only] be obtained having a complex sampling/counting system that is difficult to deploy and maintain in remote conditions" I believe that this statement is incorrect. The only disadvantages of two-filter detectors (capable of the lowest detection limits) are (i) that they are not readily portable (after having been installed), on account of their large size (2-3m), and (ii) and that they measure only Radon-222 (since Radon-220 is removed from the sampled air stream prior to entering the detector). 220Rn removal is necessary because their alpha counting system can't distinguish between i A , a-particles of different energy. The operation of the two-filter detectors is not complex; it is based primarily on a ZnS-photomultiplier counting system, a pair of centrifugal blowers, and a Campbell data logger. As such, power requirements are limited to around 100-120W at 240V when sampling from close to the surface. In spite of their size, these detectors weigh only around 100 kg, and can be readily deployed in challenging remote sites (from mountain-top to polar regions) or mobile platforms (such as ships). Furthermore, where network services are available they can be fully remotely controlled. Since calibration and instrumental background checks on the two-filter detectors are performed automatically (or via remote control), maintenance requirements are also minimal. In fact, a 1500 L model two-filter radon detector has been in service in Antarctica since February 2013 to current (October 2017), and the only user intervention required over this >4-year period has been to remove ice collected on the inlet tube on two occasions. Over this time the detector's calibration has remained guite stable, as has the lower limit of detection (25-30 mBg m-3). In most situations, however, we have found it prudent to replace the sensitive components of the two-filter detector's measurement head every 5 years to maintain a high sensitivity and low instrumental background."

**We clarified advantages and disadvantages in the text P3 L2-5.**

RC3.22: "P2 L28-29: Particular assumptions regarding the degree of equilibrium between radon and its progeny will also change under high humidity (or indeed foggy or hazy) conditions, and (during the summer months at this site when local emissions are significant), depending on the height above ground at which sampling is conducted."

**We modified the text considering that high-humidity conditions include precipitation events.**

RC3.23: "P2 L30-32: "This is a single-filter approach coupled to beta-counting and it represents, at the moment, the best compromise between detection efficiency and required resources." This claim, I feel, is somewhat misleading. As previously mentioned, two-filter detectors have low power requirements, minimal maintenance requirements, a 30 minute temporal resolution, require no assumptions to be made about the state of equilibrium between radon and its progeny, an average measurement sensitivity that rarely changes by more than 1% per year (in a roughly linear, correctable manner), and have a detection limit almost an order of magnitude better than that of the FAI PBL mixing monitor. They are, however, large (if space is an issue at the measurement location), not designed to be portable (which is only really a concern for short-term campaigns, since unpacking and initial setup can take 2 days), and they are not capable of monitoring activity concentrations of thoron progeny, or cosmogenic radionuclides. In summary, there are some advantages to using the PBL mixing monitor rather than a two-filter detector in some situations, but I think these relate more to its portability and ability to distinguish between different progeny than to resource requirements (e.g. maintenance and power)."

We clarified earlier the advantage of distinguish the different progenies and we specify at this point that the best compromise is referred to our logistics

RC3.24: "Interestingly, in their comparison of advantages/disadvantages between direct and indirect measurements, the authors fail to mention the apparent difficulty in obtaining consistent absolute radon activity concentrations from the instrument used in this study. Following claims that the instrument is readily deployable in remote environments, and that it requires minimal maintenance/resources, later (on page 5) the authors go on to say "Considering the logistic restrictions of the study site, routine quality check and sampling efficiency assessments were not possible." Problems, apparently specific to this campaign, that have prevented the authors from reporting of absolute radon concentrations in this study. However, despite the established history in the literature of applying the FAI PBL mixing monitor for atmospheric radon sampling (and other similar single-filter  $\beta$ -radiation detectors of this kind, such as the OPSIS SM200 stability monitor), few of the published studies report calibrated (absolute) radon activity concentrations. It would certainly improve the utility of these devices for applications like the one described in this study if absolute calibration of the observations was routinely possible."

We are planning to define routinely calibration of this kind of instrument and we are studying how to solve logistics and hardware problems. The main challenge is represented by the impossibility to have permanent personnel at the station during the whole campaign length.

RC3.25: "P3 L7: Regarding Figure 1b, this figure would be more useful to the reader if the view were "zoomed out" a little more. If the figure was changed such that the width represented 150-200 km, instead of about 50 km, then it would put the site in better context regarding the trajectory analysis and local influences, and would not lose too much of the local topographic detail."

We added a new Figure 1. The study site is on the left since the open sea is in front of the site and no information are present.

RC3.26: "Section 2.1: since this study is not the first application of the FAI PBL monitor, please include only the detail and theory in this section that (i) has not already been published, and (ii) pertains to the unique features of the detector operation for this study (which, as I understand, is the increased temporal resolution of sampling). Perhaps all of the detail in this section is required (the authors would be the best judge), but if other publications summarise the theory of operation (as much as it is similar to the FAI PBL mixing monitors with the slower temporal response), then it would be sufficient to refer the reader to other published works for an overview of the theory or principle of operation. This may leave more room for a more detailed analysis of the observations later."

The algorithm presented here is all original. The theory is necessary for the reader in order to appreciate the introduction of the third radioactive component while the FAI PBL provides a 2-component output. The FAI algorithm is efficient at lower latitudes where the near-constant progeny is negligible.

RC3.27: "P6 L10-12: Can the authors provide any indication of how "good" the remote soil moisture estimates are? Was there any ground-truthing performed (either for this study or in the literature)? A reference to a study where the technique has been evaluated would be sufficient if nothing specific was tested in this study.

We added suggest Brocca et al. (2017) as reference.

RC3.28: "P6 L16: Could the authors comment briefly on the results of the comparisons of trajectory calculations between 500 and 1000m that led them to their final choice?"

We modified Fig.4 and commented the picture in text using the residence time of air masses at different altitudes. The interest on 1000m altitude is caused by the impact of the inversion layer on the mixing between lower and upper air circulation in the fiord system. Furthermore, there are no large differences between residence time over Svalbard estimated for 500 and 1000 m altitudes.

RC3.29: "P6 L22-23: "The evolution of the three radioactive components (Fig. 2a) seemed to be produced by the overlapping of different sources and processes." This may well be the case, but little evidence to support this statement is provided in Figure 2a. Modelled local radon flux and air temperature are provided as companion series to the activity measurements, but there appears to be little in the way of direct consistent correlations between either of these two parameters and the more significant of the reported concentration variations in the measured activities. Perhaps including time series of wind speed, wind direction, ratios (e.g. between S:L, S:C, L:C), or trajectory-modelled time-over-land for each sample over the past 5 days would provide more information about factors contributing to the observed variability? Regarding Figure 2, please rethink the scale of the x-axis, consider decimal days or something similar. There appears to be little relationship between the axis tick marks and labels. This makes it hard to relate them to the data."

We updated Fig. 2 and we included the requested information in Fig. 3 and Fig.4

RC3.30: "P6 L24-28: Various analyses are mentioned here, but there is no evidence of them in the figures (i.e. before/after plots showing the effect of what has been achieved, and why it was necessary)."

The seasonal separation (high and low emanation periods is highlighted by grey vertical lines. The first stated analysis consisted in separately treating the two periods. We modified Fig.2 plotting black dots for the raw data and we overlapped a trend with coloured lines.

**List of major changes**

- We revised the whole paper in order to improve the English writing.
- We prepared new Figure 1
- We prepared new Figure 2
- We prepared new Figure 3
- We prepared new Figure 4
- We modified the text of section 1 in order to fix point outlined by all the reviewers
- We clarified the rationale of equations 5 to 9 fixing numbering and some sentences
- We tried to support the conversion of activities in order to compare our results to literature, even if equilibrium and detector efficiency are still critical issues. We used bold assumptions and we evidenced this aspect using \* for activities and the suffix "eq" for nuclides.
- We expressed, in section 3, activities in terms of mBq\* in order to obtain absolute values
- We updated section 3.2 with the new figures and commented them in the text
- We updated and corrected the reference list.

**High-time resolved radon-progeny measurements in the Arctic region (Svalbard Islands, Norway): results and potentialities**

Roberto Salzano1, Antonello Pasini2, Antonietta Ianniello2, Mauro Mazzola3, Rita Traversi4, Roberto Udisti4

5 1Institute for Atmospheric Pollution Research, National Research Council of Italy, Sesto Fiorentino (FI), Italy 2Institute for Atmospheric Pollution Research, National Research Council of Italy, Monterotondo (RM), Italy 3Institute of Atmospheric Sciences and Climate, National Research Council of Italy, Bologna (BO), Italy 4Department of Chemistry "Ugo Schiff", University of Florence, Sesto Fiorentino (FI), Italy

Correspondence to: Roberto Salzano (roberto.salzano@cnr.it)

- 10 Abstract. The estimation of radon progeny in the Arctic region represents a scientific challenge due to the required low limit of detection in consideration of the limited radon emanation associated with permafrost dynamics. This preliminary study highlighted, for the first time above 70°N, the possibility to monitor radon progeny in the Arctic region with a higher time resolution. The composition of the radon progeny offered the opportunity to identify air masses dominated by long-range transport, in presence or not of near-constant radon progeny instead of long and short lived progenies. Furthermore, the
- 15 different ratio between radon and thoron progenies evidenced the contributions of local emissions and atmospheric stability. Two different emanation periods were defined in accordance to the permafrost dynamics at the ground and several accumulation windows were recognized coherently to the meteo-climatic conditions occurring at the study site.

**1** Introduction**

- The detection Monitoring the levels of radionuclides within the Arctic environment is an important tool to help understanding the pathways for radionuclide transport to, within and from the Arctic (Chun, 2014; AMAP, 2010). Naturally--oOccurring radionuclideNatural Occurring Radioactive Materials (NORMs)s, emitted by are nuclides strictly related to geologic sources and associated with cosmogenic processes, which can describe air-masses origin and residence time (Baskaran, 2016). This is a key information for studying the fate of pollutants in the Arctic region, which is controlled by the meteo-climatic conditions occurring in the different seasons and in the different days of the year (Baskaran and Shaw, 2001). From a seasonal point of
- 25 view, the extension of the so-called "Arctic front" (Stohl, 2006) can deflect, in fact, air masses originated in continental areas (such as Northern Europe, Russia, Asia and North America) to higher altitudes, reducing the contribute to the deposition processes. Radon (222Rn and 220Rn have half-lives, respectively, of about 3.8 days and 55 s)Radon\_and its progeny represent an important tracer of the meteo-climatic conditions occurring in the lower atmosphere. The use of naturally-occurring nuclidesNORMs, and in particular of radon, for pollution purposes was extensively investigated at lower latitudes (Duenas et al. 2006).
- al., 1996; Perrino et al., 2001; Sesana et al., 2003; Chambers et al., 2011; 2015), especially in urban settings. These case studies

support the scientific community to use 222Rn as a comparatively simple and economical approach for defining the stability conditions of the lower troposphere and for estimating the mixing height (Pasini and Ameli, 2003; Sesana et al., 2003; Veleva et al., 2010; Griffith et al., 2013; Pasini et al., 2014; Salzano et al., 2016). The low emissive conditions of the ground, controlled by the permafrost dynamics, limit the application of this approach in polar regionsPolar Regions. The expected radon activities

- 5 in the air (we refer to especially to 222Rn that is more frequently estimated in literature) ranges in the Arctic between -30 mBq m-3, with persistent polar winds, and more than up to 400 mBq m-3 when continental air masses reached higher latitudes (Samuelson et al., 1986). This value is, of course, influenced by the latitude, by the meteo-climatic conditions, by the altitude of the sampling site and, finally, by the distance of continental areas. This rangeequirement is coherent with Antarctica (Chambers et al 2014) when "oceanic" air masses occurs and it is less stringent when "continental" or "local" air messes are
- incoming (Chambers et al., 2014). The occurrence of a melting season coupled with the higher extension of bared local and remote soils, potential sources of radon emissions, let the requirement of. The requirement of significant low levels of detection (LLD) less stringent in Svalbard islands. The approach to tThe reduction of LLD can, in fact, be based on by passed only having high-volume sampling and/or high-sensitivity detectors (Chambers et al., 2014). The available techniques can be classified considering the half-life (t1/2) of the considered isotopes. A first methodology is based on the direct
   meassurement measurement of radon nuclides (222Rn or 220Rn) detecting the in-equilibrium progeny (218Po, 214Po and 216Po, 216Po, 216Po, 216Po, 216Po, 216Po, 216Po, 216Po, 216Po, 216Po,
- with  $t_{1/2} < 205 \text{ min}$ . Some ;); the indirect techniques includebased on the detection of short-lived isotopes (such as 214Bi and 214Pb, with  $t_{1/2} < 1 \text{ hour}$ ) and of long-lived nuclides (such as 212Bi, 212Pb with  $1 < t_{1/2} < 10 \text{ hours}$ ). Finally, some ;); and the indindirect indirect methods are based on the detection of near-constant progeny (such as 210Pb and 210Bi, with  $t_{1/2} > 10^{10}$ Bi, with  $t_{1/2} > 10^{10}$ Bi and 210Bi, with  $t_{1/2} > 10^{10}$ Bi and 210Bi and 210Bi

[revised manuscript text omitted]
 naturally-occurring nuclides<del>NORMs, Eq. NORMs, Eq.</del> (1) can be described by the sum of  $\beta$  emissions produced by 222Rn progeny (Sβ), 220Rn progeny (Lβ) and some near-constant nuclides including cosmogenic isotopes (Cβ).

**$T_{\beta} = [{}^{214}Pb]_{\beta} \frac{S_{\sharp}}{S_{\sharp}} + [{}^{214}Bi]_{\beta} \frac{S_{\sharp}}{S_{\sharp}} + [{}^{212}Pb]_{\beta} + [{}^{212}Bi]_{\beta} \frac{L_{\sharp}L_{\sharp}}{L_{\sharp}} + C_{\beta}$**

(5)

Excluding from C $\beta$  the contribution of 210Pb, due to the low  $\beta$  energy emission ( $E_{\beta} < 100 \text{ keV}$ ) where the detector has a very low efficiency (Lee and Burgess, 2014), the remaining near-constant nuclides are 210Bi ( $t_{1/2} \sim 5 \text{ days}$  and  $E_{\beta} \sim 1162 \text{ keV}$ ), 10Be  $(t_{1/2} > 10^6 \text{ years and } E_{\beta} \sim 556 \text{ keV})$  and 14C  $(t_{1/2} \sim 5700 \text{ years and } E_{\beta} \sim 156 \text{ keV})$ . While the 14C contribution is 5 limited by the low efficiency of detectors at low energies and by the limited amount of carbon present on filters (below 1 µg m-3), the 10Be component is limited by the low activities present in the atmosphere. Summarizing, we have one rapid-decay component ( $S_{\beta}$  decreases 60 - 70 % within one hour) and one near-constant member ( $C_{\beta}$ ). The intermediate term ( $L_{\beta}$ ) reduces its activity to about 5 - 15 % after one hour. The mixing between those three components defines the final decay behavior 10 observable at an hourly scale with four different counting steps. We can have two different seasonal behaviors in the Arctic region. The first: one: one occurs ring occurring especially during the Arctic winter, when the local emission of radon (both 222Rn and 220Rn) is negligible ( $L_{\beta} \simeq 0$ ) and the residence time of aerosol over sea (more than 2 days) is higher in presence of the so-called "Arctic haze" ( $C_{\beta} > 0 - \frac{210}{2} = \frac{210}{Bi_{\beta}}$ ). The second;); one occursringoccurring especially in the summer, when the local component is significant and the Arctic haze is reduced  $(S_{\beta} \gg L_{\beta} > C_{\beta})$ . We assume under both conditions that transient equilibrium is occurring between the two progenies ( $[^{214}Pb]_{air} = [^{214}Bi]_{air}$  and  $-[^{212}Pb]_{air} = [^{212}Bi]_{air}$ ). Some 15 bias can occur especially during the summer when the local source is dominating over transport and the disequilibrium between progenies can be significant.

The Bateman's solutions support the development of the above mentioned above. The above mentioned differential equation Eq. (2), concerning the sampling phase, can be solved using the Bateman's solutions, considering the two seasonal assumptions:

| $[^{214}Bi]_{S}]_{\mathcal{F}} = 1.51[^{214}Pb]_{S\mathcal{F}}$ | (6a) |
|-----------------------------------------------------------------|------|
| $[^{214}Pb]_{air} = 1.97 \nu^{-1} [^{214}Pb]_{s}$               | (6b) |

$$[^{212}Bi]_{S}]_{\cancel{F}} = 1.02[^{212}Pb]_{S}]_{\cancel{F}}$$
(6c)

25
$$[^{212}Pb]_{air} = 1.03 \nu^{-1} [^{212}Pb]_{S}]_{\mathcal{F}}$$
 (6d)
 $C_{\beta}^{air} [^{210}Bi]_{air} = \nu^{-1} [^{210}Bi} \nu^{-1} C_{\beta} ]_{\mathcal{F}}$  (6e)

The Eq. (4) regarding the counting intervals must be solved for each period. The solution must consider the first and the last counting periods: from 0 to 10 minutes (first interval  $T_{\beta}^{1}$ ) and from 40 to 50 minutes (forth interval  $T_{\beta}^{4}$ ) after the end of the air

30 sampling.

$$T_{\beta}^{1} = \epsilon_{1024} [^{214}Pb]_{S}]_{\mathcal{F}} d_{1}^{1} + \epsilon_{3272} [^{214}Bi]_{S}]_{\mathcal{F}} d_{2}^{1} + \epsilon_{570} [^{212}Pb]_{S}]_{\mathcal{F}} d_{3}^{1} + \epsilon_{2252} [^{212}Bi]_{\mathcal{F}S} d_{4}^{1} + C_{\beta}$$
(7a)
* * *
**(7b)**

The coefficients in Eq. (7a and 7b) are the detector efficiencies at each energy ( $\epsilon_{keV}$ ) and the decay parameters ( $d_i^n$ ) obtained solving exponential equations (Eq. 4) for each ith isotope at each nth counting interval. We were not able to determine routinely

- 5 the detector efficiency at each energy but it was possible to make some experiments with a similar instrument and some reference materials such as a KCl standard (we prepared a known 40K filter with Eβ~ 1311 keV) and a 137Cs-contaminated soil (we prepared a 137Cs-enriched filter with Eβ~ 531 keV where the activity was determined by γ-spectrometry). This preliminary calibration requires a stronger effort for estimating precisely the efficiency at different energies but a relative ratio between the detector efficiency at 570, 1024 and 2252 keV normalized to the efficiency at 1024 keV was estimated in order to study the variations of the three β-emitting components (Sβ, Lβ and Cβ). We found that ε570 ~ 0.41ε1024, ε1162 ~ 1.1ε1024,
- $\epsilon_{2252} \sim 1.8\epsilon_{1024}$  and  $\epsilon_{3272} \sim 2.1\epsilon_{1024}$ . Substituting these parameters to Eq. (7a and 7b) and solving the system including a  $^{220}$ Rn to  $^{222}$ Rn ratio (*f*), we obtained:

$$S_{\beta} = 1.97 \frac{(N_{\beta,1} - N_{\beta,4})}{(2.15 + 0.74f)} v^{-1}$$
(89a)

$$L_{\beta} = 3.74f \frac{(N_{\beta,1} - N_{\beta,4})}{(2.15 + 0.74f)} v^{-1}$$
(89b)

15
$$C_{\beta} = [N_{\beta,4} \frac{(2.14f+1.89)}{(2.15+0.74f)} (N_{\beta,1} - N_{\beta,4})] v^{-1}$$
 (89c)

- Awhere both all-ll quantities are expressed in cps m-3. Nevertheless, we prefer to enhance the contribution to the scientific community assuming the equilibrium between progenies during the observed period (and indicate  $^{214}Pb_{eq} = S_{B_{-}}^{212}Pb_{eq} = S_{B_{-}}^{212}Pb_{eq}$  $L_{\beta}$  2222. Assuming also that  $en\epsilon_{1024} \sim 10\%$ , having in mind that further efforts are is necessary for a reliable calibration, s is necessary for a reliable calibration, the comparison of our results with literature is possible obtaining activities expressed in mBq m-3. Regarding CThis cannot be done for  $\beta$ , the conversion requires -a deeper knowledge about this component and we 20 prefer consequently to keep relative counts. since it is necessary to The minimizestimation of the three components was obtained ation of where both quantities are expressed in cps m-3. The estimation of the three components was obtained minimizing the chi-squared indicator, calculated between the four counting intervals and the respective values simulated between the two end-member situations (ing tills incetilities is necessary to a deeper knowledge about this component.  $\mathcal{C}_{\mathcal{C}}$ 0 or  $\underline{\mathbb{C}}_{\beta} = 0$ ) structure of the estimation of the three components. and 4th counting steps  $C_{\beta} = 0$  and  $L_{\beta} = 0$ ). The optimization 25 algorithm was developed in the R-Project programming environment (R Core Team, 2016)counting steps.). The lower limit of detection, in terms of 222Rn, of the stability monitor was estimated at 150 mBq<del>0.0.15 Bq</del> m-3 (Salzano et al, 2016) using an independent technique.). Considering the logistic restrictions of the study site, routine quality check and sampling efficiency assessments were not possible. These limitations forced us to express methods the three components in terms of relative
- 30 radioactivity (cps m-3)  $^{214}Pb_{eq}$  and  $^{212}Pb_{eq}$  as mBq\* m-3 and to estimate formation of CB as relative counts. The respective

lowest limit of detections were 200<del>indicated as</del>  $mBq*m^{-3}$ , with a LLD of 0.0035,250 0.013 $mBq*m^{-3}$  and 0.0<del>072</del>15 cps m- 3with allowest limit of detections were indicated as LLD of -3, respectively.

**2.2 Soil Rn-flux**

[revised manuscript text omitted]

The dynamics of the inversion layer in the fiord system

- 25 will be investigated in the next future considering additional observations that can describe the fiord system better than a coarse-resolution model like HYSPLIT. The evolution of absolute humidity provided sSome confirmations about the importance of this process can be obtained from the evolution of absolute humidity (Fig.4be). The general increasing trend over the whole campaign was, in fact, -combined to accumulation windows, probably related to atmospheric stability, and to abrupt decrements (more than 10-g m-3) that were probably associated with glacier flows. These slope flows, with a katabative
- 30 behavior indicated also, were by coupled to high wind speed (more than 8 m s-1), and they were coincident with significant  $C_{\beta}$  activities.

[revised manuscript text omitted]